



# DeerLab: A comprehensive toolbox for analyzing dipolar EPR spectroscopy data

Luis Fábregas Ibáñez[1], Gunnar Jeschke[1], and Stefan Stoll[2]

[1]ETH Zurich, Laboratory of Physical Chemistry, Vladimir-Prelog-Weg 2, 8093 Zurich, Switzerland
[2]University of Washington, Department of Chemistry, Seattle, WA 98195, USA

**Correspondence:** Stefan Stoll (stst@uw.edu)

**Abstract.** Dipolar EPR spectroscopy (DEER and other techniques) enables the structural characterization of macromolecular and biological systems by measurement of distance distributions between unpaired electrons on a nanometer scale. The inference of these distributions from the measured signals is challenging due to the ill-posed nature of the inverse problem. Existing analysis tools are scattered over several applications with specialized graphical user interfaces. This renders comparison, reproducibility, and method development difficult. To remedy this situation, we present DeerLab, an open-source MATLAB toolbox for analyzing dipolar EPR data that is modular and implements a wide range of methods. We show that DeerLab can perform one-step analysis based on separable non-linear least squares, fit dipolar multi-pathway models to multi-pulse DEER data, and run global analysis with parameter-free distributions. Important aspects of uncertainty analysis are discussed as well.

## 1 Introduction

Dipolar electron paramagnetic resonance (EPR) spectroscopy encompasses a growing family of techniques for determining distributions of nanometer-scale distances between unpaired electrons. These distance distributions provide valuable information for the structural characterization of macromolecular or biological systems that is complementary to information obtained by other techniques. For structurally disordered or highly complex systems, where established techniques may fail, such distance distributions provide unique information. The family of dipolar EPR spectroscopy techniques includes double electron–electron resonance (DEER) (Milov et al., 1981, 1984; Pannier et al., 2000b), double quantum coherence (DQC) (Saxena and Freed, 1996, 1997; Borbat et al., 2013), relaxation-induced dipolar modulation enhancement (RIDME) (Kulik et al., 2001; Milikisyants et al., 2009), single-frequency technique for refocusing (SIFTER) (Jeschke et al., 2000), and several other related techniques (Borbat et al., 2013; Di Valentin et al., 2014; Hintze et al., 2016; Pribitzer et al., 2017; Borbat and Freed, 2017; Doll and Jeschke, 2017; Milikisiyants et al., 2018). All of them provide a time-domain signal that depends on the dipolar interaction between pairs of electrons. From this time-domain signal, the distance distribution is inferred.

Due to its mathematical nature, the robust inference of distance distributions from noisy dipolar EPR spectroscopy data is not straightforward. Many approaches have been proposed to tackle this problem (Pannier et al., 2000a; Jeschke et al., 2002; Bowman et al., 2004; Jeschke et al., 2004; Chiang et al., 2005b, a; Jeschke et al., 2006; Sen et al., 2007; Brandon et al., 2012; Stein et al., 2015; Dzuba, 2016; Matveeva et al., 2017; Srivastava and Freed, 2017; Edwards and Stoll, 2016; Rein et al., 2018;





Timofeev et al., 2018; Worswick et al., 2018; Hustedt et al., 2018; Edwards and Stoll, 2018; Fábregas Ibáñez and Jeschke, 2019, 2020), each with its pros and cons. Some of these methods have found widespread use, via software packages such as DeerAnalysis (Jeschke et al., 2006), GLADD/DD (Brandon et al., 2012), and LongDistances (Altenbach, 2019).

However, there are several major challenges with the current situation. (i) A comparative assessment of the relative merits of various methods is missing. (ii) Many methods have been argued based on anecdotal evidence from small datasets, and their performance has not been assessed comprehensively. (iii) Reproducibility of analysis results is very limited due to the lack of common platforms for data sharing and data analysis.

To remedy this situation, we introduce DeerLab. It is an open-source software for data analysis in dipolar EPR. It is based on MATLAB (MathWorks, 2020) and consists of a collection of modular functions, analogous to EasySpin (Stoll and Schweiger, 2006) and Spinach (Hogben et al., 2011). This has several distinct advantages over a graphical user interface (GUI). (i) It allows for very flexible workflow designs, easily adapting to different experimental situations. (ii) All existing methods can be directly compared on a single platform. (iii) Automation and processing of large datasets becomes straightforward. (iv) Scripted data analysis improves reproducibility and collaboration. (v) It provides a foundation for implementing new methodologies. (vi) It can be embedded into other software, such as tools for protein structure modelling based on distance distributions (Jeschke, 2018). The disadvantage is an accessibility barrier for potential users without programming skills. This disadvantage can be remedied by building a dedicated GUI for standard workflows as an additional software layer.

This paper is structured as follows. We start by summarizing the theoretical basics of dipolar EPR spectroscopy. Then, we illustrate the functionality of DeerLab through a series of examples. First, we show how to reproduce well-established workflows such as Tikhonov regularization and multi-Gauss fits. We then demonstrate how DeerLab can perform one-step analysis based on separable non-linear least-squares optimization, fit dipolar multi-pathway models to multi-pulse DEER data, and run global analysis with parameter-free distributions. Finally, several sections are dedicated to important aspects of uncertainty analysis. The DeerLab scripts for all figures are available in the Supporting Information.

## 2 Theoretical basics

This section summarizes the central theoretical concepts of dipolar EPR spectroscopy that DeerLab is based on. For more details, see (Jeschke, 2012) and (Jeschke, 2016). The theory is limited to $S = 1/2$ spins with isotropic $g$ values, without any orientation selection, at most two spins per protein, no exchange coupling, weak dipolar coupling, and no conformer-dependent relaxation rates.

Dipolar EPR spectroscopy techniques measure the dipole-dipole couplings between spins via the modulation of the amplitude $V(t)$ of a spin echo as a function of the position $t$ of one or more pump pulses. The echo amplitude is modeled as (Milov et al., 1981)

$$V(t) = V_0 \cdot V_\text{intra}(t) \cdot V_\text{inter}(t) \tag{1}$$

where $V_0$ is the echo amplitude in the absence of any pump pulses, $V_\text{intra}$ describes the pump-pulse-induced modulation due intra-molecular dipolar couplings, and $V_\text{inter}$ describes the modulation due to intermolecular couplings. $V_0$ is a constant





prefactor that we set to one from now on in order not to complicate the notation unnecessarily. DeerLab takes $V_0$ into account as a fitting parameter.

The product of intra- and inter-molecular dipolar modulations can be written in a general form as

$$V(t) = \int\limits_0^\infty K(t,r) P(r) \, \mathrm{d}r \tag{2}$$

$P(r)$ is the distribution of intra-molecular spin-spin distances $r$ on the protein or other complex, normalized such that $\int_0^\infty P(r)\mathrm{d}r = 1$.

    $K(t,r)$ is the kernel that captures how the complete dipolar modulation is determined by the distance distribution. It includes
the inter-molecular modulation (Fábregas Ibáñez and Jeschke, 2020). For standard 4-pulse DEER it is

$$K(t,r) = \left[(1-\lambda) + \lambda K_0(t,r)\right] B(t,\lambda) \tag{3}$$

Here, $\lambda$ is the modulation depth. $K_0$ is the elementary kernel

$$K_0(t,r) = \int\limits_0^1 \cos\left[(1 - 3\cos^2\theta)Dr^{-3}t\right] \mathrm{d}\cos\theta \tag{4}$$

with the dipolar coupling constant

$$D = \frac{\mu_0}{4\pi} \frac{g_\mathrm{e}^2 \mu_\mathrm{B}^2}{\hbar} \tag{5}$$

where $g_\mathrm{e}$ is the $g$-value of the free electron, $\mu_\mathrm{B}$ the Bohr magneton, $\mu_0$ the magnetic constant, and $\hbar$ is the reduced Planck constant. $K_0$ assumes full orientation averaging and unlimited excitation bandwidth. The subscript $0$ distinguishes this elementary kernel from more general kernels such as Eq. (3).

    $B(t,\lambda)$ represents the inter-molecular modulation and is commonly called the background. It can be modeled as a stretched-
exponential function

$$B(t,\lambda) = \exp\left(-\kappa_d \lambda |t|^{d/3}\right) \tag{6}$$

where $\kappa_d$ is a decay rate constant and $d$ is the dimensionality (Kutsovsky et al., 1990; Milov et al., 1998; Jeschke et al., 2002). Other background models are possible (Kattnig et al., 2013).

    Experimentally, the echo amplitude is measured only for a discrete set of usually equally spaced time points $t_i$, yielding a
dipolar signal vector $\boldsymbol{V}$ with $n$ elements $V(t_i)$. For numerical analysis, $P(r)$ is represented as a discrete distance distribution vector $\boldsymbol{P}$ with $m$ elements $P(r_j)$ at equally spaced $r_j$. With this, Eq. (2) reads

$$\boldsymbol{V} = \boldsymbol{K}\boldsymbol{P} \tag{7}$$

where $\boldsymbol{K}$ is the $n \times m$ kernel matrix with elements $(\boldsymbol{K})_{ij} = K(t_i, r_j)\Delta r$, and $\Delta r$ is the increment in the distance domain.





Experimental data $\boldsymbol{V}_{\mathrm{exp}}$ deviate from in Eq. (1) due to presence of noise. From experiments, it was found that the noise

distribution in DEER signals is well approximated by an uncorrelated Gaussian distribution with zero mean and constant

variance (Edwards and Stoll, 2016):

$$\boldsymbol{V}_{\mathrm{exp}} = \boldsymbol{V} + \mathcal{N}(\boldsymbol{0}, \sigma^2 \boldsymbol{I}) \tag{8}$$

Inferring the distance distribution from the dipolar signal formally requires inversion of the kernel matrix

$$\boldsymbol{P} = \boldsymbol{K}^{-1} \boldsymbol{V}_{\mathrm{exp}} \tag{9}$$

However, $\boldsymbol{K}$ is generally badly ill-conditioned (it has an extremely large condition number). This renders the inverse problem

ill-posed, and the results obtained by Eq. (9) are highly unstable, erratic, and unreliable, especially in the presence of noise.

Because of the ill-posedness, inferring distance distributions from the dipolar signals poses a major challenge in dipolar EPR

spectroscopy data analysis.

## 3   Current approaches

Currently, two families of methods are commonly used in dipolar EPR spectroscopy data analysis. They differ in whether the

distance distribution is represented as a parametric model or as a parameter-free model. Both methods stabilize the solution

and are widely used since they are simple and often effective.

The use of these established methods is very easy with DeerLab. Fig. 1 shows a short DeerLab script that performs a full

comparative analysis of experimental data using parameter-free distributions (Tikhonov regularization) and parametric models

(Gauss and worm-like chain). In the following, we discuss these two families of methods from the perspective of DeerLab.

### 3.1   Parameter-free distributions

If $\boldsymbol{P}$ is represented as a parameter-free vector, regularization methods are used to determine the solution. If background and

modulation depth are known and fixed, the associated regularized optimization problem has the form

$$\boldsymbol{P}_{\mathrm{fit}} = \operatorname*{argmin}_{\boldsymbol{P} \geq 0} \left( \|\boldsymbol{V}_{\mathrm{exp}} - \boldsymbol{K}\boldsymbol{P}\|^2 + \alpha^2 \mathcal{R}(\boldsymbol{L}\boldsymbol{P}) \right) \tag{10}$$

The first term represents the sum of squared residuals, i.e. $\chi^2$ without normalization by the noise variance, which we assume

constant across the signal (see Eq. (8)). It quantifies quality of the fit of the model to the data. The second term is an additional

penalty term that, together with the non-negativity constraint $\boldsymbol{P} \geq 0$, stabilizes the solution. The smoothing regularization

matrix $\boldsymbol{L}$ is a numerical approximation of a differential operator to impose smoothness, and $\alpha$ is the regularization parameter,

which controls the balance between data agreement and regularization.

Regularization methods differ in the choice of the penalty norm $\mathcal{R}$. Tikhonov regularization (Tikhonov, 1963) was the first

regularization approach introduced for dipolar data analysis (Bowman et al., 2004; Chiang et al., 2005a; Jeschke et al., 2004).





```
1  % Load experimental dataset
2  [t,V] = deerload('experiment.DTA');
3
4  % Pre-processing
5  V = correctphase(V);
6  t = correctzerotime(V,t);
7  r = linspace(1,7,400);
8
9  % Fit with Tikhonov regularization
10 [Vfit1,Pfit1] = fitsignal(V,t,r,'P',@bg_exp,@ex_4pdeer);
11
12 % Fit a Gaussian parametric model
13 [Vfit2,Pfit2] = fitsignal(V,t,r,@dd_gauss2,@bg_exp,@ex_4pdeer);
14
15 % Fit a parametric chain model
16 [Vfit3,Pfit3] = fitsignal(V,t,r,@dd_wormchain,@bg_exp,@ex_4pdeer);
17
18 % Compare the fits
19 plot(r,Pfit1,r,Pfit2,r,Pfit3)
```

**Figure 1.** Basic DeerLab script for processing experimental data. The data is first pre-processed, and then distance distributions are fitted using three different approaches: Tikhonov regularization, a two-Gaussian model, and a worm-like chain model.

Other approaches such as total variation (TV), Huber regularization, and Osher's Bregman-iterated regularization are beneficial in certain cases (Fábregas Ibáñez and Jeschke, 2019).

In the absence of the non-negativity constraint, the problem in Eq. (10) could be solved directly by $\boldsymbol{P}_{\text{fit}} = \bar{\boldsymbol{K}} \boldsymbol{V}_{\text{exp}}$, with the
$\alpha$-dependent regularized pseudoinverse of the kernel matrix,

$$\bar{\boldsymbol{K}} = (\boldsymbol{K}^{\text{T}} \boldsymbol{K} + \alpha^2 \boldsymbol{L}^{\text{T}} \boldsymbol{L})^{-1} \boldsymbol{K}^{\text{T}} \tag{11}$$

in the case of Tikhonov regularization. However, due to the non-negativity constraint, this is not possible, and Eq. (10) is solved using non-negative least-squares optimization algorithms (Lawson and Hanson, 1974; Bro and Jong, 1997; Chen and Plemmons, 2009). $\bar{\boldsymbol{K}}$ will be useful for uncertainty analysis (see Section 9).

The selection of $\alpha$ has been optimized for a large testset in previous work (Edwards and Stoll, 2018) and replicated (Fábregas Ibáñez and Jeschke, 2019), revealing that selection methods such as the Akaike information criterion (AIC) (Akaike, 1974) or general cross validation (GCV) (Golub et al., 1979) can be superior to L-curve criteria (Hansen, 2000; Chiang et al., 2005a; Jeschke et al., 2006) if Gaussian white noise is the only source of error.

DeerLab provides a flexible regularization framework including all the aforementioned $\alpha$-selection, non-iterated and iterated
regularization methods, as well as a wide selection of solvers. In particular, implementation of Huber and TV regularization is much improved compared to our original study (Fábregas Ibáñez and Jeschke, 2019), such that these now perform similarly to Tikhonov regularization, invalidating the conclusions on the under-performance of Huber and TV regularization from that initial work.

Fig. 2 shows how regularization approaches can be compared using DeerLab. Fig. 2a presents an example of low-noise
dipolar signal processed via Tikhonov, TV and Osher's Bregman (Tikhonov) iterated regularization. All methods yield similar





results. Fig. 2b illustrates how regularization methods perform in the presence of strong noise. In such cases more differences arise between the different methods.

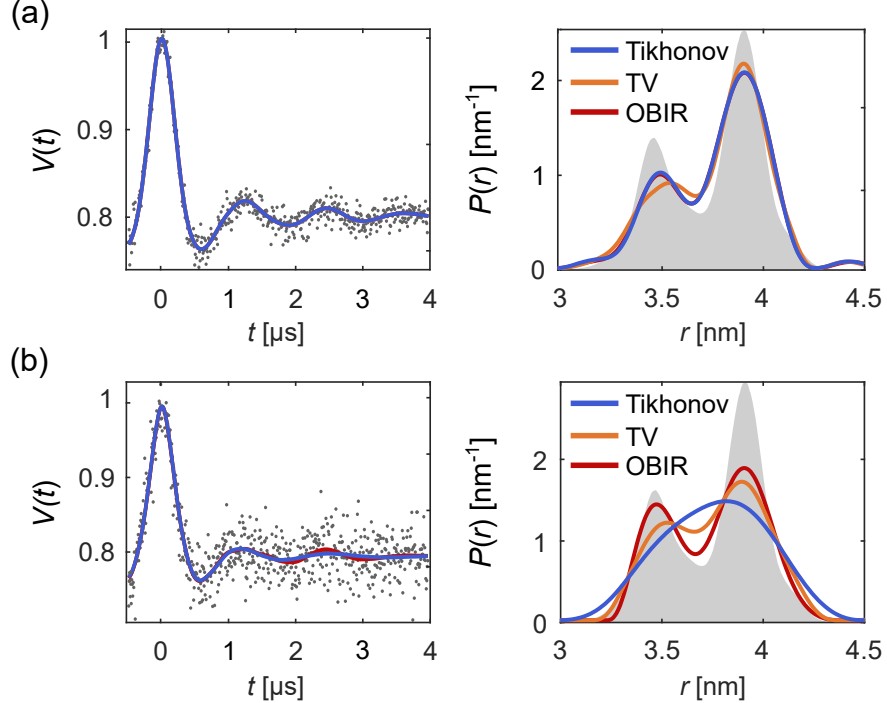

**Figure 2.** Analysis of background-free dipolar signals with regularization. A distance distribution is fitted to a low-noise (a) and a high-noise (b) signal using Tikhonov regularization (blue), TV regularization (orange), and Osher's Bregman iterated (OBIR) regularization (red). In all cases, the AIC was used for the selection of the regularization parameter. The input data (simulated) are shown as grey dots and the ground truth distance distribution is shown as grey shaded areas.

The outcomes of regularization analysis depend strongly on the choice of penalty norm, regularization operator, and $\alpha$. For the remainder of this work, if not specified otherwise, we will use the Tikhonov penalty equipped with the second-order difference operator $\boldsymbol{L}_2$ and the AIC for $\alpha$-selection.

### 3.2 Parametric models

In an alternative representation, $\boldsymbol{P}$ is described as a parametric model $\boldsymbol{P}[\boldsymbol{\theta}]$ with $(\boldsymbol{P}[\boldsymbol{\theta}])_i = P(t_i, \boldsymbol{\theta})$, where $\boldsymbol{\theta}$ is a vector of a small number of parameters. This is fitted to the data using

$$\boldsymbol{\theta}_{\text{fit}} = \underset{\boldsymbol{\theta}}{\text{argmin}} \|\boldsymbol{V}_{\text{exp}} - \boldsymbol{K}\boldsymbol{P}[\boldsymbol{\theta}]\|^2 \tag{12}$$

The reduced dimensionality of the $\boldsymbol{\theta}$-space compared to $\boldsymbol{P}$-space often stabilizes the solution of the ill-posed inverse problem to a sufficient extent, without the need of regularization.





While bimodal Gaussian distributions were the first parametric models (Pannier et al., 2000a), the idea was generalized to a linear combination of $N$ Gaussian distributions (Sen et al., 2007) (which we will refer to as multi-Gauss model)

$$\boldsymbol{P}[\boldsymbol{\theta}] = \sum_{i=1}^{N} a_i \boldsymbol{p}_i[\overline{r}_i, \sigma_i] \tag{13}$$

where $a_i$ are the amplitudes and $\boldsymbol{p}_i$ are the normalized Gaussian basis functions parameterized by their center distances $\overline{r}_i$ and widths $\sigma_i$. Other parametrizations of the amplitudes can be used (Brandon et al., 2012; Stein et al., 2015).

To determine the optimal number $N$ of Gaussians in the multi-Gauss model, Sen et al. proposed a statistical F-test (Sen et al., 2007), while Hustedt and coworkers (Stein et al., 2015; Hustedt et al., 2018) introduced the corrected Akaike information criterion (AICc) (Sugiura, 1978; Hurvich and Tsai, 1989) and the Bayesian information criterion (BIC) (Schwarz, 1978). Two

more recent approaches utilize Monte-Carlo simulations to determine the optimal multi-Gauss model (Dzuba, 2016; Timofeev et al., 2018). Parametric models are not limited to Gaussian basis functions. Many other basis function types (or mixtures thereof) can be employed, e.g. 3D-Rice distributions (Domingo Köhler et al., 2011), spherical distributions (Ionita et al., 2008; Kattnig and Hinderberger, 2013), random-coil models (Fitzkee and Rose, 2004), or worm-like chain models (Wilhelm and Frey, 1996).

In a milestone for parametric modelling, Brandon et al. expanded the use of parametric models to include the modulation depth and a stretched-exponential background into the analysis of the signal in their software GLADD/DD (Brandon et al., 2012; Stein et al., 2015). This results in a time-domain parametric model

$$\boldsymbol{V}[\boldsymbol{\theta}] = \boldsymbol{V}[\lambda, \boldsymbol{\theta}_P, \boldsymbol{\theta}_B] = \boldsymbol{K}[\lambda, \boldsymbol{\theta}_B]\boldsymbol{P}[\boldsymbol{\theta}_P] \tag{14}$$

In this, the parameter vector $\boldsymbol{\theta}$ not only includes the distance distribution parameters $\boldsymbol{\theta}_P$, but also the modulation depth $\lambda$ and

the background parameters $\boldsymbol{\theta}_B$. In general, a parametric time-domain model can be fitted to the experimental data by solving

$$\boldsymbol{\theta}_{\text{fit}} = \underset{\boldsymbol{\theta}}{\operatorname{argmin}} \|\boldsymbol{V}_{\text{exp}} - \boldsymbol{V}[\boldsymbol{\theta}]\|^2 \tag{15}$$

DeerLab expands upon this, allowing the design and fitting of any kind of time-domain or distance distribution parametric model. Automated multi-Gauss fitting and model selection using AIC, BIC, and other metrics are provided as well. In Fig. 3 we provide such an example of multi-Gauss fitting with DeerLab. The signal is fitted using the time-domain model Eq. (14)

and a varying number of Gaussians as the distance distribution model. Model selection based on the AIC optimizes the number of Gaussians, with a decent fit of the distance distribution.

It is, however, crucially important to keep in mind that approaches based on parametric models may suffer from selection bias, i.e. the bias introduced by limiting the analysis to a specific family of models and by using a particular criterion for model selection within that family (Freedman, 1983; Lukacs et al., 2009). Parametric model fits may also be affected by confirmation

bias, i.e. the tendency to process data in a way that matches one's preconceptions and avoids contradiction of prior belief (Nickerson, 1998). In contrast, regularization approaches use parameter-free distributions and are less affected by these biases. Therefore, it is recommended to use parametric models only when there are strong reasons to prefer them over parameter-free models. Even then, the results should always be contrasted, and presented along with a parameter-free analysis.





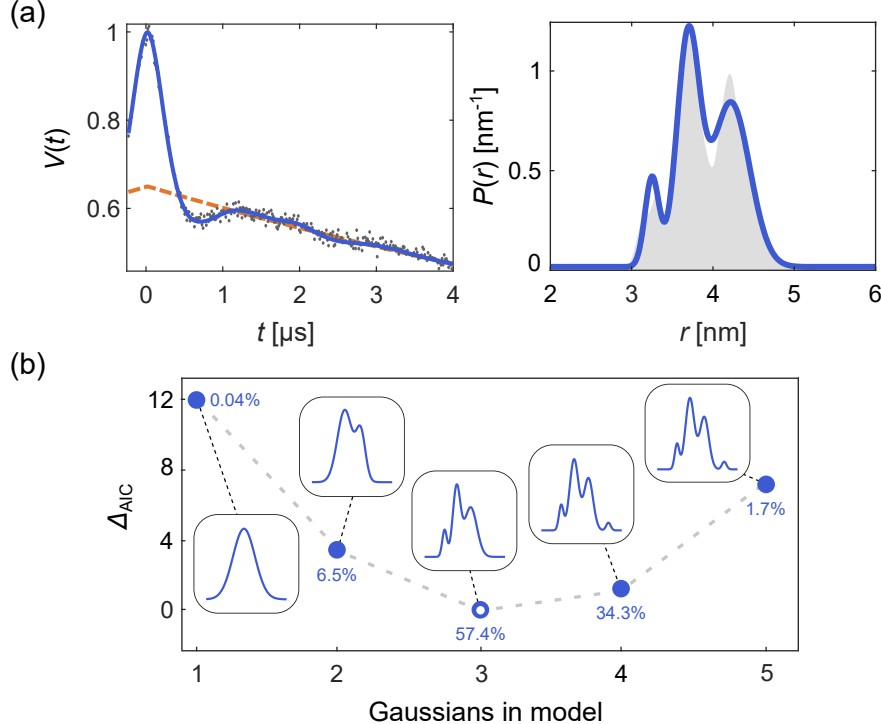

**Figure 3.** Time-domain multi-Gauss fitting of a simulated 4-pulse DEER signal. (a) The data are given as grey dots, the ground truth distance distribution as a shaded area, and the corresponding fits of a 3-Gauss model as blue lines. The background fit is given as dashed orange line. (b) The difference in AIC values as a function of the number of Gaussians is shown in blue, and the corresponding fits are given in the insets. The differences $\Delta_{\mathrm{AIC}}$ are relative to the model with the lowest AIC value. The corresponding Akaike weights (see Section 10) are given next to each model. The model with the lowest AIC value (i.e. largest Akaike weight) is selected as the optimal model.

## 4   One-step analysis

Neither parametric model fitting nor regularization are ideal. Parametric models can be fit directly to the full time-domain signal, but are strongly limited by how well they can approximate the distribution ground truth. On the other hand, regularization yields parameter-free distance distributions that accommodate a much larger range of ground truths, but regularization cannot directly include time-domain parameters such as the background and the modulation depth.

With regularization methods, it is therefore common practice to use a two-step approach: (1) fit a parametric background
model to the time-domain signal and correct the signal by the fitted background (either by division or subtraction), and (2) apply regularization to the background-corrected signal. Software based on this approach includes DeerAnalysis (Jeschke et al., 2006), LongDistances (Altenbach, 2019). Including the fitted background into the kernel for step (2), as in Eq. (3), does also not eliminate the need for the two-step approach (Fábregas Ibáñez and Jeschke, 2020).





The two-step analysis is sub-optimal, because the background fit in step (1) relies on the assumption that the oscillation periods in the time-domain signal are much shorter than the overall signal length. Many experimentally observed signals do not satisfy this assumption. Therefore, the two-step analysis cannot robustly process these types of signals.

The most desirable approach is to simultaneously fit both the time-domain parameters $\boldsymbol{\theta}$ and a parameter-free distance distribution $\boldsymbol{P}$ to the time-domain signal $\boldsymbol{V}_{\text{exp}}$ in one step, i.e.

$$(\boldsymbol{\theta}_{\text{fit}}, \boldsymbol{P}_{\text{fit}}) = \underset{\boldsymbol{\theta}, \boldsymbol{P} \geq 0}{\operatorname{argmin}} F(\boldsymbol{\theta}, \boldsymbol{P}) \tag{16}$$

with the regularized objective function

$$F(\boldsymbol{\theta}, \boldsymbol{P}) = \|\boldsymbol{V}_{\text{exp}} - \boldsymbol{K}[\boldsymbol{\theta}]\boldsymbol{P}\|^2 + \alpha^2 \mathcal{R}(\boldsymbol{L}\boldsymbol{P}) \tag{17}$$

Eq. (16) can be solved using a variety of constrainable non-linear optimization algorithms by combining $\boldsymbol{\theta}$ and $\boldsymbol{P}$ into a single parameter vector and applying all the constraints for $\boldsymbol{\theta}$ and $\boldsymbol{P}$ (Altenbach, 2019). However, this does not take full advantage of the special structure of the problem, i.e. that it is a penalized least-squares problem with a model for $\boldsymbol{V}$ that is linear in $\boldsymbol{P}$ and non-linear in $\boldsymbol{\theta}$.

To take advantage of this structure, DeerLab implements a nested subspace optimization algorithm based on separable non-linear least-squares (Budil et al., 1996; Golub and Pereyra, 2003; Sima and Van Huffel, 2007). It separates the $\boldsymbol{\theta}$ and $\boldsymbol{P}$ spaces and uses a non-linear least-squares algorithm to solve

$$\boldsymbol{\theta}_{\text{fit}} = \underset{\boldsymbol{\theta}}{\operatorname{argmin}} F(\boldsymbol{\theta}, \boldsymbol{P}[\boldsymbol{\theta}]) \tag{18}$$

where $\boldsymbol{P}[\boldsymbol{\theta}]$ is the optimal parameter-free distance distribution for a given $\boldsymbol{\theta}$, determined via regularization

$$\boldsymbol{P}[\boldsymbol{\theta}] = \underset{\boldsymbol{P}' \geq 0}{\operatorname{argmin}} F(\boldsymbol{\theta}, \boldsymbol{P}') \tag{19}$$

as in Eq. (10), using a dedicated non-negative linear least-squares algorithm. (Note that $\boldsymbol{P}$ is now a parametric model since it depends on $\boldsymbol{\theta}$, although this includes only modulation depth and background parameters.) For solving Eq. (18), the regularization term in $F$ is neglected. Once $\boldsymbol{\theta}_{\text{fit}}$ is obtained, $\boldsymbol{P}_{\text{fit}}$ is calculated as $\boldsymbol{P}[\boldsymbol{\theta}_{\text{fit}}]$. The algorithm is iterative and is illustrated in Fig. 4. It starts with an initial guess for $\boldsymbol{\theta}$ and determines the associated $\boldsymbol{P}$ from Eq. (19). This $\boldsymbol{P}$ is then used by the algorithm of Eq. (18) to determine the next $\boldsymbol{\theta}$, which is then used again by the algorithm in Eq. (19) to get the next $\boldsymbol{P}[\boldsymbol{\theta}]$, and so on until convergence is reached. The optimization in Eq. (19) can utilize any form of regularization, and it can be run with a fixed regularization parameter $\alpha$ or optimize it each time.

Fig. 5 shows an example that compares this one-step approach to the traditional two-step analysis, using progressively more truncated dipolar signals. For a sufficiently long signal (Fig. 5a) both approaches yield similar results, as the signal is long enough for the oscillations to average out and decay, facilitating the separate fit of the background in the two-step analysis. If the signal is truncated as in Fig. 5b-d, the two-step analysis cannot properly fit the background anymore. In the most truncated case (Fig. 5d), the two-step analysis does not even identify any major modulated component (a fitted $\lambda \approx 0$) fitting the data





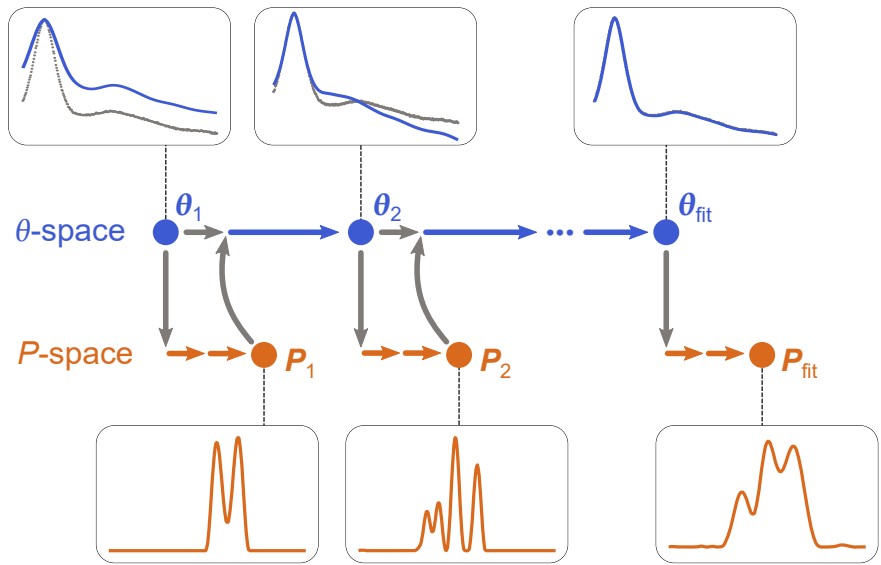

**Figure 4.** One-step analsyis of dipolar signals using separable non-linear least-squares optimization. On the upper level (blue), a set of parameters $\boldsymbol{\theta}_n$ are computed by optimization of Eq. (18). For each $\boldsymbol{\theta}_n$, a corresponding distance distribution $\boldsymbol{P}_n$ is computed by optimization of Eq. (19). This procedure is repeated until a minimum of the objective function is found for an optimal parameter set $\boldsymbol{\theta}_{\mathrm{fit}}$ and leading to the corresponding optimal distribution $\boldsymbol{P}_{\mathrm{fit}}$.

mostly as a background signal, and cannot fit any reasonable distribution to it. In contrast, one-step analysis correctly identifies
the background in all cases and recovers the underlying distance distribution with reasonable fidelity.

This shows that DeerLab's ability to simultaneously fit the background and a parameter-free distribution opens up the possibility of fitting parameter-free distance distributions to signals which might have been deemed unanalyzable in the past.

## 5  Multi-pathway models

Dipolar EPR spectroscopy found widespread use with 4-pulse DEER. Since then, experimental dipolar EPR spectroscopy has
developed into a set of diverse techniques, with signals exhibiting a variety of features. However, the data processing has not evolved much from its 4-pulse DEER origins.

The key operator in the analysis is the dipolar kernel $K(t,r)$ (see Eq. 2). It represents the experiment, i.e. it describes how the time-domain signal is obtained from a given distance distribution. If the kernel cannot account for a feature in the signal, it is because the kernel model is incomplete. In these cases, it is preferable to improve the kernel rather than to tweak the signal
into an artificial 4-pulse DEER signal.

One example of this is RIDME, where it is known that the measured signals in $S > 1/2$ systems contain overtones not present in 4-pulse DEER (Razzaghi et al., 2014). If disregarded, these overtones cause distortions in the distance distribution if the 4-pulse DEER kernel is used to analyze the data. This can be avoided by including the overtones into the model (Keller





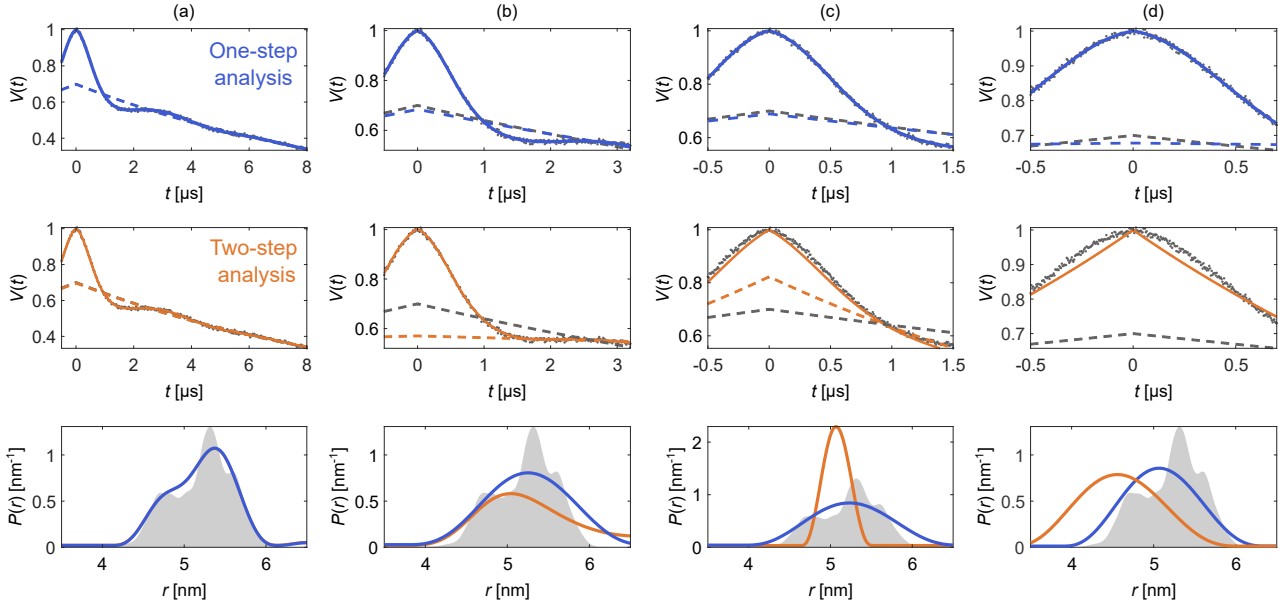

**Figure 5.** Comparison of one-step and two-step analysis. A stretched-exponential background and a parameter-free distance distribution were fitted to a simulated 4-pulse DEER signal. The analysis was done by either fitting both simultaneously (blue) or by fitting the background followed by the distribution (two-step analysis, orange). The analysis was repeated on the same signal with increasing truncation (a-d). The two-step analysis fails in case (d). The data is given as grey dots, and the fitted signal and background are given as solid and dashed lines, respectively. The true background is given as a grey dashed line for reference. The parameter-free distance distributions obtained via Tikhonov regularization are given as respectively colored lines and the ground truth as a shaded area.

et al., 2017). In DeerLab, one can include a set of overtones with a background into the kernel model, which can be used to
directly fit primary RIDME data via, e.g. regularization.

Other examples are multi-pulse DEER sequences, which generally are not fully modeled. All multi-pulse DEER experiments feature modulations in addition to the basic modulation of Eq. (3). Despite their clear dipolar origin, these are regarded as undesirable "artifacts": the "2+1 artifact" in 4-pulse DEER (Jeschke, 2012; Teucher and Bordignon, 2018), and "artifacts" or "residues" in 5-pulse and 7-pulse DEER (Borbat et al., 2013; Spindler et al., 2015; Breitgoff et al., 2017b, a; Doll and Jeschke,
2017; Milikisiyants et al., 2018). Several experimental and processing approaches have been published that aim to remove these contributions from the total signal to recover the idealized dipolar evolution function (Borbat et al., 2013; Spindler et al., 2015; Teucher and Bordignon, 2018; Breitgoff et al., 2017b, a). These approaches introduce further experimental or theoretical complexity.

However, these additional contributions are actual dipolar signals. They may even provide strong oscillations at times when
the oscillations from the main signal have decayed, thus contributing to the signal-to-noise ratio of the experiment. Instead of removing these contributions due to the lack of a proper model, it is advantageous to extend the model to explicitly include these contributions.



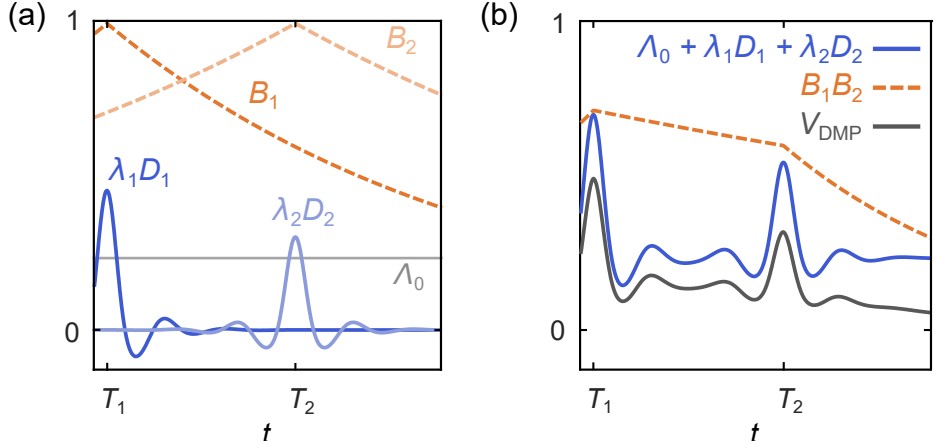

**Figure 6.** Schematic representation of a dipolar multi-pathway DEER signal. (a) Background decays (orange dashed lines) and dipolar evolution functions (blue solid lines) are shown for two different dipolar pathways. The unmodulated component $\Lambda_0$ is given as a solid grey line. (b) The overall signal (black) is given by the sum over all pathway dipolar evolution functions (blue) times the product over all pathway background decays (orange).

DeerLab includes such an extended model for multi-pulse DEER, derivable from spin density matrix dynamics. In this model, which we call the dipolar multi-pathway model, the overall signal is a combination of several dipolar signals arising from dipolar pathways of varying amplitudes and refocusing times (Salikhov et al., 1981; Kutsovsky et al., 1990; Kattnig et al., 2013; Borbat et al., 2013; Spindler et al., 2015). The total signal is given by Eq. (2) with the general kernel

$$K(t,r) = \left[ \Lambda_0 + \sum_{p=1}^{N} \lambda_p K_0(n_p(t - T_p), r) \right] \cdot \prod_{p=1}^{N} B(n_p(t - T_p), \lambda_p) \tag{20}$$

where $\Lambda_0$ is the total contribution of the unmodulated dipolar pathways, the index $p$ runs over all $N$ modulating dipolar pathways, $T_p$ are the refocusing times of the individual modulated dipolar pathways, $\lambda_p$ are the amplitudes of the modulated dipolar pathways, $n_p$ are the harmonics of the individual modulated pathways, and the background function is as in Eq. (6). A schematic representation is shown in Fig. 6. The kernel for standard 4-pulse DEER from Eq. (3) is a special case of Eq. (20) with $N = 1$, $T_1 = 0$, $n_1 = 1$ $\Lambda_0 = 1 - \lambda$, and $\lambda_1 = \lambda$.

Discretization of the kernel in Eq. (20) gives the same expression as in Eq. (7),

$$V = K[\theta]P \tag{21}$$

with the parameter set $\theta$ comprising $\Lambda_0$, all $\lambda_p$, $n_p$, $T_p$, as well as $\kappa_d$, and $d$.

Fig. 7 shows examples of how DeerLab can analyze multi-pulse DEER data in terms of a multi-pathway model, thus alleviating the need for experimental correction protocols or signal pre-processing (beyond phase correction). Note that experimental schemes which generate multiple datasets with some pathways shifted with respect to each other (Breitgoff et al., 2017a) can profit from global analysis (vide infra) to stabilize the accurate estimation of pathway parameters.





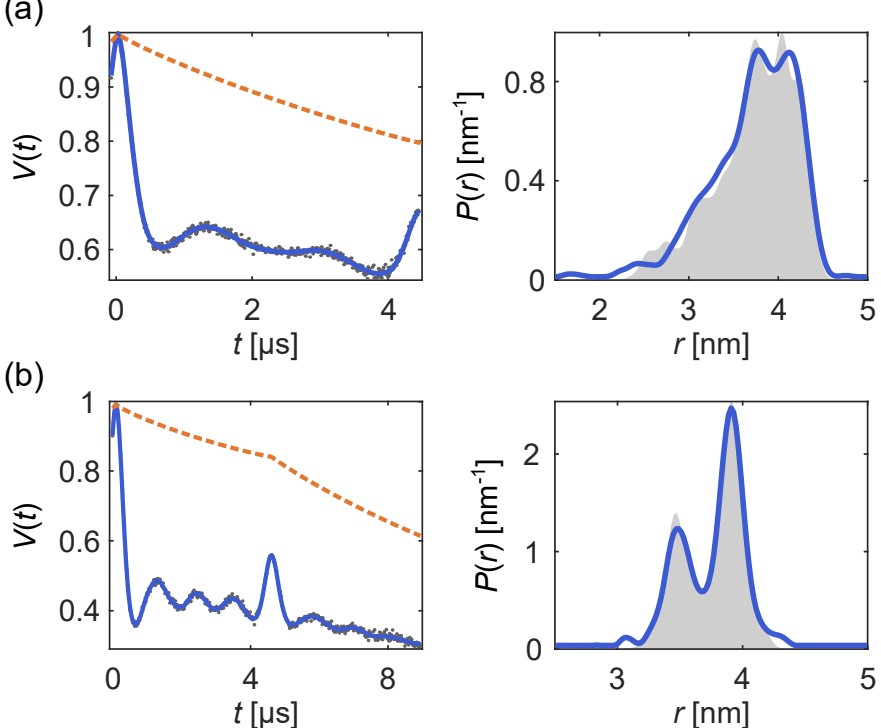

**Figure 7.** One-step analysis of multi-pulse DEER signals with the dipolar multi-pathway model for (a) 4-pulse DEER with the "2+1 artifact" and (b) 5-pulse DEER with its 4-pulse "artifact". All data were normalized to the zero-time of the maximal contribution and fitted with two modulated dipolar pathways. The data are given as grey dots, the signal and distribution fits are given as solid blue lines, the background fit is given as an orange dashed line, and the ground truth is given as a shaded area.

This analysis of multi-pulse DEER signals shows the benefits of using full models for dipolar signals instead of removing or avoiding "artifacts" to try to match partial models. DeerLab provides a compact framework for testing, developing, and applying more complete models.

## 6 Global analysis

Global analysis denotes the situation where a single model is fit simultaneously to multiple data sets. In dipolar EPR, this was first used for fitting a multi-Gauss distribution model to several DEER datasets (Brandon et al., 2012). Later, Tikhonov regularization was employed to fit a short and a long DEER trace simultaneously (Rein et al., 2018).

Global analysis is implemented in DeerLab for both parametric and non-parametric distance distribution models, and for an arbitrary number of dipolar signals. The global optimization problem of a set of $M$ dipolar signals $\boldsymbol{V}_{\exp,i}$ using a model





depending on parameters $\boldsymbol{\theta}$ and on $N$ parameter-free distance distributions $\boldsymbol{P}_j$ is

$$(\boldsymbol{\theta}_{\text{fit}}, \{\boldsymbol{P}_{\text{fit}}\}) = \underset{\boldsymbol{\theta}, \{\boldsymbol{P}\} \geq 0}{\operatorname{argmin}} [F(\boldsymbol{\theta}, \{\boldsymbol{P}\}) + G(\{\boldsymbol{P}\})] \tag{22}$$

with $\{\boldsymbol{P}\} = \{\boldsymbol{P}_1, \ldots, \boldsymbol{P}_N\}$ and

$$F(\boldsymbol{\theta}, \{\boldsymbol{P}\}) = \sum_{i=1}^{M} w_i \frac{\|\boldsymbol{V}_{\text{exp},i} - \boldsymbol{V}_i[\boldsymbol{\theta}, \{\boldsymbol{P}\}]\|^2}{\sigma_i^2} \tag{23}$$

$$G(\{\boldsymbol{P}\}) = \alpha^2 \sum_{j=1}^{N} \mathcal{R}(\boldsymbol{L}\boldsymbol{P}_j) \tag{24}$$

with a similar expression without $\{\boldsymbol{P}\}$ and $G(\{\boldsymbol{P}\})$ for a fully parametric model. $\sigma_i$ are the noise levels of the individual signals. The parameter vector $\boldsymbol{\theta}$ includes the parameters needed to generate all signals $\boldsymbol{V}_i$, where each $\boldsymbol{V}_i$ typically depends only on a subset of the parameters.

The quantities $w_i$ in Eq. (23) are the weights that determine the contribution of each signal to the objective function. The default is $w_i = 1$, meaning that each data point from each signal contributes equally, given its noise level, to the objective function. Different values of $w_i$ can be used to indicate preferential weighing.

If all signals $\boldsymbol{V}_i$ derive from a single distance distribution, then we can use Eq. (22) together with

$$\boldsymbol{V}_i[\boldsymbol{\theta}, \boldsymbol{P}] = \boldsymbol{K}_i[\boldsymbol{\theta}]\boldsymbol{P} \qquad \text{or} \qquad \boldsymbol{V}_i[\boldsymbol{\theta}] = \boldsymbol{K}_i[\boldsymbol{\theta}]\boldsymbol{P}[\boldsymbol{\theta}] \tag{25}$$

Here, each $\boldsymbol{K}_i$ describes a different experiment (different pulse sequence, different trace length, etc.) and depends on a subset of the parameters in $\boldsymbol{\theta}$. The distance distribution $\boldsymbol{P}$ can be either parametric (in which case $\boldsymbol{\theta}$ includes the distribution parameters), or it can be parameter-free. In the latter case, Eq. (22) is solved using separable non-linear least squares. As an example, in Fig. 8 we simultaneously fit a 4-pulse DEER signal and a 5-pulse DEER signal with its secondary 4-pulse pathway contribution, using a model with a single parameter-free distribution, but separate backgrounds and modulation depths for the two signals. The distance distribution underlying both signals is nicely recovered.

Note, however, that the analysis of dipolar signals of different length or obtained under different dynamical decoupling conditions may be inconsistent if different conformers have different dephasing rates (Baber et al., 2015).

Another common global analysis situation is when the measured signals stem from several samples containing a mixture of chemical or structural components, each with its own distinct distribution, and related to each other via additional conditions such as a chemical equilibrium. The simplest case is a system equilibrated between two forms A and B (A $\rightleftharpoons$ B), e.g. a protein–ligand binding equilibrium or a monomer–dimer equilibrium. In this case, the total distributions are

$$\boldsymbol{P}_i = x_{\text{A},i}\boldsymbol{P}_{\text{A}} + x_{\text{B},i}\boldsymbol{P}_{\text{B}} \tag{26}$$

with the component distributions $\boldsymbol{P}_{\text{A}}$ and $\boldsymbol{P}_{\text{B}}$. The mole fractions $x_{\text{A},i}$ and $x_{\text{B},i}$ depend on the location of the equilibrium, which might vary among the samples via ligand concentration, matrix composition, and other effectors. Either the mole fractions or the underlying equilibrium constant can be included among the fitting parameters.





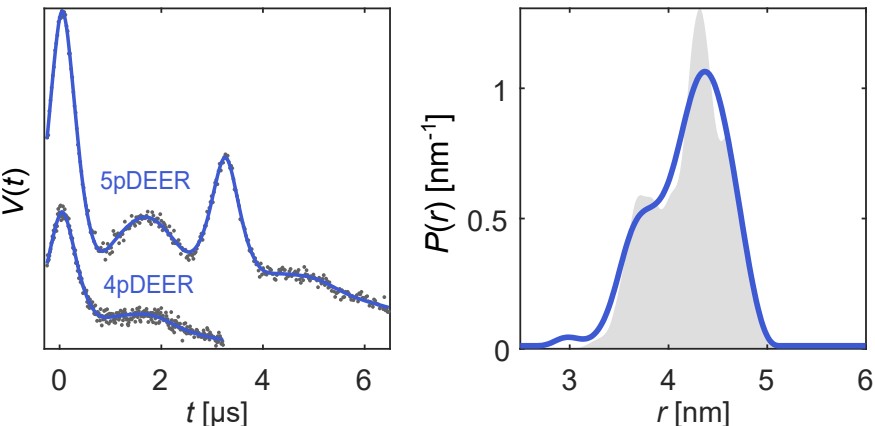

**Figure 8.** Global analysis of a 4-pulse DEER and a 5-pulse DEER signal with its secondary 4-pulse pathway contribution, both derived from the same distance distribution. The simulated data are given as grey dots, the ground truth distribution is given as a shaded area, and the signal and distribution fits are given as solid blue lines.

Such titration or dose–response datasets have been analyzed using multi-Gaussian distribution models for the component distributions (Stein et al., 2015; Martens et al., 2016; Collauto et al., 2017; Barth et al., 2018; Jagessar et al., 2020). As discussed above, however, parameter-free distributions may often be a preferable choice. DeerLab enables global fitting of an arbitrary number of data sets with regularization approaches and, thus, analysis of titration datasets in terms of parameter-free distributions. As an example, Fig. 9 shows such an analysis of a protein–ligand binding assay, using signals with different noise levels, trace lengths, backgrounds, and modulation depths (Fig. 9a) with a model that includes the bound-protein mole fractions among the parameters. The global analysis gives good fits to the time-domain data and results in two parameter-free distributions that capture the underlying ground truth well (Fig. 9b). The extracted mole fractions are consistent with the underlying binding equilibrium (Fig. 9c). Alternatively, one can skip the separate determination of the mole fractions and include the dissociation constant directly as a parameter in the global analysis of the dipolar data. In summary, this example shows that DeerLab allows global analysis of titration datasets using parameter-free distributions.

## 7 Global and local minima

Even when a fitted model agrees with the experimental data, and a minimum of the objective function has been located, it does not mean that the only or the best fit has been found. The objective function can have multiple minima, meaning that the located minimum is not necessarily the global minimum (see Fig. 10). Which minimum is found depends mainly on where the search is started. The boundaries set on the parameter space can also influence the outcome if a minimum is located outside these boundaries. Other factors, e.g. the numerical algorithms and convergence parameters used for the optimization, can affect this as well.





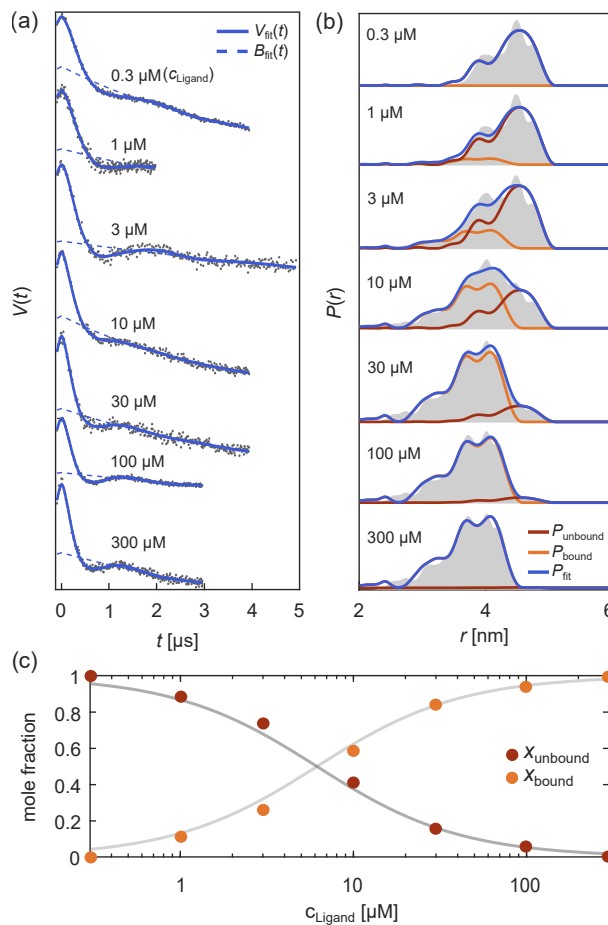

**Figure 9.** Global fitting of titration data of a protein–ligand binding equilibrium with parameter-free distributions. In (a) 4-pulse DEER traces with different trace lengths, modulation depths, backgrounds, and noise levels were simulated for different ligand concentrations added to a protein concentration of $1\,\mu$M. These parameters as well as the mole fractions of bound/unbound protein for each trace and two parameter-free distance distributions for the bound and unbound states (via Tikhonov regularization) were fitted simultaneously. The simulated data are given as grey dots and the fitted signals and backgrounds are given as solid and dashed blue lines, respectively. In (b) the distance distribution fits for the unbound (red) and bound (orange) states are given as well as the combined fitted distribution (black) for the different ligand concentrations. The ground truth sum distributions are given as grey shaded areas. In (c) the fitted mole fractions of the unbound (blue) and bound (orange) states are given as colored dots. The solid grey lines represent the ground truth for a dissociation constant of $K_\mathrm{D} = 5.65\,\mu$M.

While there are dedicated global optimization algorithms, the simplest approach to find a global minimum is to repeat the optimization process with different starting values in order to explore the parameter space more fully. After sampling enough starting points, a set of minima is found and the one with the lowest objective function value is taken as the global minimum (see Fig. 10).





While these procedures can be costly, it is recommended to routinely check that variation of algorithmic parameters does not yield a lower minimum.

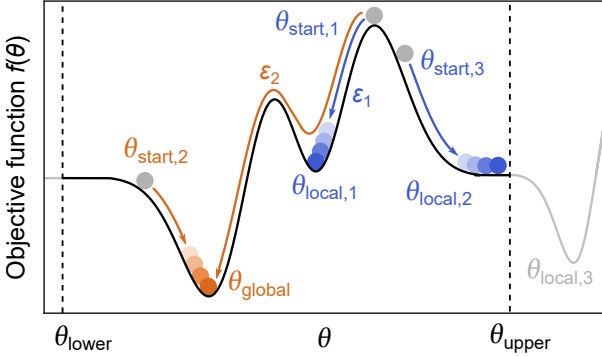

**Figure 10.** Global vs. local minima. During optimization of a parameter $\theta$, by minimization of an objective function $f(\theta)$ (black line), several local minima (blue) might be found instead of the global minimum (orange). The global minimum can be found by varying the starting point, the lower/upper boundaries $\theta_{\mathrm{upper}}$ and $\theta_{\mathrm{lower}}$ (to find minima which might be outside the bounds, e.g. $\theta_{\mathrm{local},3}$), or the convergence parameters $\varepsilon$ of the numerical solver such that some local minima might be ignored.

## 8   Goodness of fit

After obtaining a fitted model, an important step is to assess the goodness of the fit. If the fit is not good, either the optimization failed to locate an appropriate minimum (see previous section), or the model is inapplicable (e.g. oversimplified) for the given
experimental data and a better model needs to be used.

To assess the goodness of fit, several procedures can be utilized. A direct test is to compare the histogram of the normalized residuals $|V_{\mathrm{exp},i} - V_i[\boldsymbol{\theta}]|/\sigma$ to the standard Gaussian distribution $\mathcal{N}(0,1)$. Here, $\sigma$ is the noise standard deviation estimated either from a dataset with several, individually stored scans (Edwards and Stoll, 2016), from the standard deviation of the (flat) imaginary part of the signal, or from the residuals of the signal minus the fitted model. Accuracy of the estimate of $\sigma$
decreases in the sequence indicated, due to a possible imbalance in quadrature channels and the slight model inadequacy that is unavoidable for ill-posed problems. An example of this is shown in Fig. 11b. If the histogram deviates strongly from a Gaussian distribution, the fit is considered inadequate. Alternatively, the comparison can also be based on a statistical test.

Another method is to examine the reduced $\chi^2$ value

$$\chi_{\mathrm{red}}^2 = \frac{1}{N_{\mathrm{dof}}} \frac{\|V_{\mathrm{exp}} - V[\boldsymbol{\theta}_{\mathrm{fit}}]\|^2}{\sigma^2} \tag{27}$$

Here, $N_{\mathrm{dof}}$ is the number of degrees of freedom, which can be taken as approximately equal to the number of data points minus the number of model parameters. A good fit is characterized by $\chi_{\mathrm{red}}^2 \approx 1$. Note that the use of $\chi_{\mathrm{red}}^2$ for non-linear models is not rigorous. Also, the notion of (effective) number of parameters for a parameter-free distribution model, estimated as $\mathrm{tr}(\boldsymbol{K}\bar{\boldsymbol{K}})$ with $\bar{\boldsymbol{K}}$ defined in Eq. (11), is not straightforward (Edwards and Stoll, 2016).





Additional methods for assessing goodness of fit are discussed in Budil et al. (1996) and (Hansen et al., 2012).

## 9 Uncertainty analysis

Up to this point, we took advantage of knowing the ground truth when we assessed quality of the solutions. However, in experimental work the ground truth is unknown. The presence of noise in experimental signals introduces uncertainty about the underlying noise-free dipolar signal and results in uncertainty about the model parameters. This, in turn, affects the strength of the conclusions that can safely be drawn. For example, $r_0 = (3.2 \pm 0.1)$ nm supports much more confident conclusions about $r_0$ than $r_0 = (3.2 \pm 0.9)$ nm. It is therefore crucial to estimate and report parameter uncertainties. Reporting fitted parameters and distance distributions extracted from experimental data without accompanying uncertainty estimates is meaningless. Several approaches have been proposed for uncertainty estimation, including validation of the regularization model (Jeschke et al., 2006; Altenbach, 2019), curvature matrices (Stein et al., 2015), and Bayesian inference (Edwards and Stoll, 2016).

DeerLab provides two separate methods for uncertainty estimation, both for model parameters $\theta_i$ and for vector elements $P_j$ of parameter-free distance distributions.

The first method estimates parameter uncertainties from the covariance matrix (Budil et al., 1996). For a fully parametric model with parameter vector $\boldsymbol{\theta}$, this is well established (Hustedt et al., 2018). The covariance matrix for $\boldsymbol{\theta}$ is

$$\boldsymbol{\Sigma_\theta} = \sigma^2 (\boldsymbol{J}^\mathrm{T} \boldsymbol{J})^{-1} \tag{28}$$

where $\boldsymbol{J_\theta}$ is the Jacobian matrix of derivatives with elements $J_{ij} = \partial V_j[\boldsymbol{\theta}]/\partial \theta_i$ evaluated at $\boldsymbol{\theta} = \boldsymbol{\theta}_\mathrm{fit}$ and is calculated using numerical differentiation.

There is also a simple way to obtain the covariance matrix $\boldsymbol{\Sigma_P}$ for the elements of a parameter-free $\boldsymbol{P}$ determined via regularization, but only if the non-negativity constraint for $\boldsymbol{P}$ is disregarded. It is obtained by propagating the time-domain noise covariance matrix $\sigma^2 \mathbf{I}$ (see Eq. (8)) to the distance domain by (Weese, 1992; Kasper et al., 2002)

$$\boldsymbol{\Sigma_P} = \sigma^2 \bar{\boldsymbol{K}} \bar{\boldsymbol{K}}^\mathrm{T} \tag{29}$$

with $\bar{\boldsymbol{K}}$ defined in Eq. (11). Note that this can be utilized even for models that depend on other parameters, since $\boldsymbol{V}$ always depends linearly on $\boldsymbol{P}$, as shown above.

From the covariance matrices in Eqs. (28) and (29), the standard errors of a parameter $\theta_i$ or a distribution vector element $P_i$ are obtained as

$$\sigma_{\theta_i} = \sqrt{(\boldsymbol{\Sigma_\theta})_{ii}} \qquad \text{and} \qquad \sigma_{P_i} = \sqrt{(\boldsymbol{\Sigma_P})_{ii}} \tag{30}$$

These are used to estimate symmetric confidence intervals (CIs) around the fitted parameter for a confidence level $\gamma$ with boundaries

$$\theta_{\mathrm{fit},i} \pm z_\gamma \sigma_{\theta_i} \qquad \text{and} \qquad P_{\mathrm{fit},i} \pm z_\gamma \sigma_{P_i} \tag{31}$$





where $z_\gamma$ is the $\gamma$-quantile of a standard Gaussian or Student's t-distribution.

This method for estimating CIs is simple and stands on a sound theoretical basis. However, it has several limitations. (1) It
approximates the parameter likelihood by a Gaussian distribution centered around the fitted $\boldsymbol{\theta}_{\text{fit}}$ and $\boldsymbol{P}_{\text{fit}}$. While this is often a
reasonable approximation, it can still fail in many cases. (2) It assumes that the parameters are unbounded. This is not fulfilled
in the analysis of dipolar signals, as most parameters are constrained to a certain range, e.g. $0 \leq \lambda \leq 1$ and $P_i \geq 0$. Although
the boundary conditions can be imposed by cropping the CIs in Eq. (31) to the parameter range (Wang, 2008), the calculation
of the CIs is still based on an unbounded assumption. Hence, the CIs do not include the additional information provided by the
constraints. (3) In the case of a model that depends on both parameters $\boldsymbol{\theta}$ and a parameter-free distribution $\boldsymbol{P}$, the covariances
between $\theta_i$ and $P_j$ are not accounted for, potentially leading to underestimation of uncertainties.

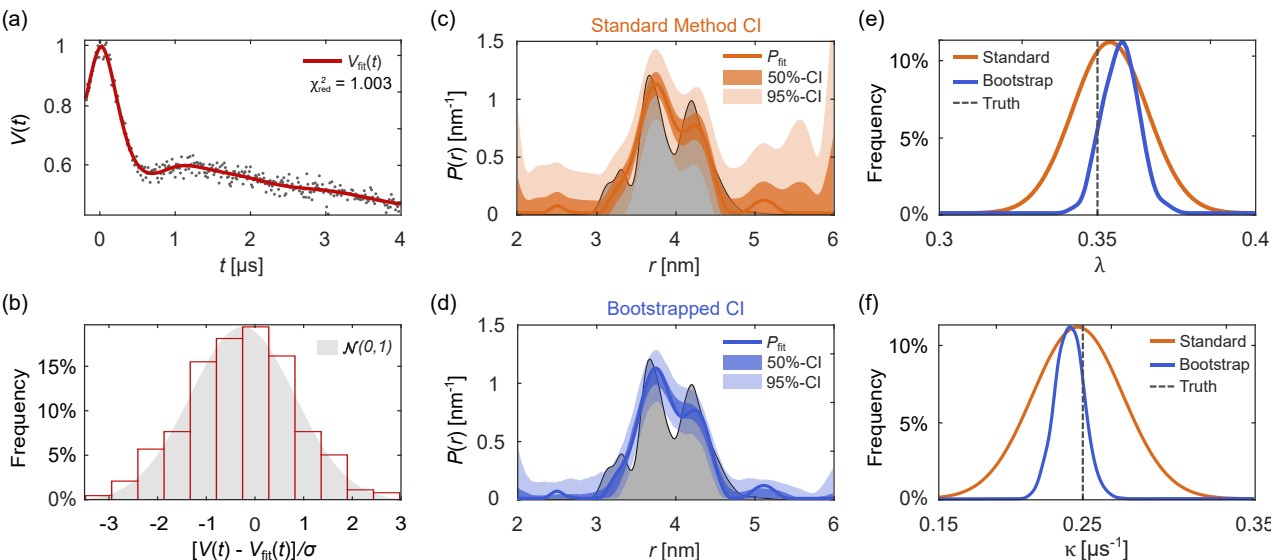

**Figure 11.** Uncertainty analysis. In (a) a simulated noisy 4-pulse DEER signal with exponential background (grey dots) is fitted with a
parameter-free distance distribution (red line). In (b), the red bars show the histogram of normalized residual values and the reference
standard Gaussian distribution as a grey shaded area. In (c) and (d) the fitted distance distribution is given as blue/orange lines as well as
the 50% and 95% confidence intervals (shaded areas) obtained via the standard method (orange, top) and via bootstrapping (blue, bottom).
The ground truth distribution is shown as a shaded grey area for reference. The estimated distribution of the fitted modulation depth $\lambda$ (e)
and background decay rate constant $\kappa$ (f) obtained via the standard method are given as orange lines. The kernel density estimation obtained
via bootstrapping are given as blue lines. The true values of $\lambda$ and $\kappa$ are given as a dashed grey line for reference. All bootstrap results were
obtained from 1000 bootstrap samples and the regularization parameter was optimized using AIC for the original signal and fixed throughout
the analysis.

A more general and accurate method for the estimation of parameter uncertainty involves bootstrapping (Efron and Tibshi-
rani, 1986; Banks et al., 2010). This is a Monte Carlo resampling method based on generated synthetic signals with different
noise realizations. The variant implemented in DeerLab first generates $N$ synthetic traces $\boldsymbol{V}_k$ by adding $N$ different noise





realizations to the fitted model $\boldsymbol{V}[\boldsymbol{\theta}_{\text{fit}}, \boldsymbol{P}_{\text{fit}}]$. The noise is drawn from a Gaussian distribution with standard deviation $\sigma$ estimated from the experimental data. Then, the $N$ bootstrap traces $\boldsymbol{V}_k$ are analyzed in the same fashion as the original dataset $\boldsymbol{V}_{\text{exp}}$, resulting in $N+1$ fitted parameter vectors $\boldsymbol{\theta}_{\text{fit},k}$ and distance distributions $\boldsymbol{P}_{\text{fit},k}$. The distributions of the parameter values and the distance distribution vector elements are then taken as approximations of the underlying parameter uncertainty distributions.

The bootstrap method, while costly, has several important advantages over the method based on covariance matrices. All information provided by parameter constraints is included in the estimation. Additionally, model-free estimations of parameter distributions are obtained, without the need to assume a Gaussian distribution. These can be statistically analyzed in multiple ways to quantify the parameter uncertainty. For example, in analogy to above, one can define the confidence interval for a parameter $\theta$ with confidence level $\gamma = 1 - \alpha$ as

$$\left( \theta_{1-\alpha/2} \, , \, \theta_{\alpha/2} \right) \tag{32}$$

where $\theta_{1-\alpha/2}$ and $\theta_{\alpha/2}$ are the $(1-\alpha/2)$-th and $(\alpha/2)$-th percentiles of the bootstrapped $\theta$ distribution.

Fig. 11 shows an example of parameter uncertainty estimation, both using the standard method and bootstrap. Although both methods lead to similar confidence intervals, the bootstrapped solution provides narrower intervals thanks to the use of the additional information provided by parameter boundaries and the non-negativity constraint $\boldsymbol{P} \geq 0$. While the standard method provides an easily accessible uncertainty estimation, the use of bootstrapping is recommended for producing final results.

## 10   Model comparison

The above uncertainty analysis is performed under the assumption of a specific model for the distance distribution (parametric or parameter-free), the background, and the experiment. All estimated uncertainties capture variability only within this assumed model. The possibility that the data could be explained equally well, or better, by other models is not incorporated.

Model selection approaches, as outlined above for parametrized and parameter-free distance distribution (selection of number of Gaussians, regularization parameter selection), provide convenient quantitative decision criteria for picking one distribution model over another in a principled fashion. On the other hand, background models are often chosen ad hoc. Finally, the choice of the standard DEER experiment model in Eq. (3) is based on physical assumptions (no orientation selection, no bandwidth limitations, no conformer-dependent phase memory times, no exchange couplings, etc.) that might not all be fully valid for a given experimental situation. Whether principled or ad hoc, any model selection eliminates model uncertainty from the analysis and leads to bias.

It is therefore preferable to compare and report the relative performance of a series of plausible models, without picking a winner. This can be accomplished using Akaike weights, defined as (Burnham and Anderson, 2003)

$$w_{\text{AIC},i} = \frac{\mathrm{e}^{-\Delta_{\text{AIC},i}/2}}{\sum_{k=1}^{M} \mathrm{e}^{-\Delta_{\text{AIC},k}/2}} \tag{33}$$





where $\Delta_{\mathrm{AIC},i}$ is the difference between the AIC value of model $i$ and the lowest AIC value within the set of models. The Akaike weights give the probability that model $i$ is the best among the set of $M$ candidate models, given the data. This can be used to compare a set of different parametric models, or to compare a series of parameter-free distribution models differing in the regularization parameter $\alpha$.

Figure 12a illustrates this for a set of parametric models differing in complexity (number of components) and type of basis function. The analysis finds Akaike weights of about 25% for the 2-Gauss, 3-Gauss and 1-skewed Gauss models, indicating that the data do not provide enough evidence to clearly identify a best model. Fig. 12b shows a comparison for parameter-free distribution models obtained with different $\alpha$ values. It is apparent that models over a range of $\alpha$ values are similarly likely. Therefore, in neither case is there a clear "winner" model. Note that uncertainty on the regularization parameter $\alpha$ could also

be propagated to the resulting distance distribution and thereby included in the confidence bands. In cases where this is not applicable and depending on the conclusions that one wants to draw, is may be prudent to list all models that fit the data reasonably well, as opposed to picking the model with the highest $w_i$.

Despite these simple numerical procedures for model comparison, researchers need to use careful judgment in which models are included in a comparison and explicitly disclose the reasoning behind all model choices. Also, if the correct model is not

included, then even such model comparison will lead to biased or incorrect conclusions. This is our rational for recommending routine comparison of results obtained with parametrized models to those obtained by parameter-free analysis, as the latter imposes the least constraints on the shape of the distribution.

## 11    Concluding remarks

Dipolar EPR spectroscopy requires reliable and robust data processing tools. The associated software should also be flexible

and adaptable to quickly incorporate new developments in the field. DeerLab collects many existing as well as several novel data analysis methods in a flexible, robust, and reliable manner.

By allowing analysis workflows to be shared, DeerLab provides a significant step towards solving reproducibility issues plaguing dipolar EPR spectroscopy analysis. With DeerLab, the analysis scripts can be provided along with published data, leading to improved reproducibility. Through an online repository (see Code Availability below), all DeerLab versions remain

available, further enhancing reproducibility.

We showed how DeerLab can serve as a powerful method development tool. It allows the discovery and testing of new processing techniques. We illustrated this by introducing several extensions: one-step analysis with parameter-free distributions, the dipolar multi-pathway model, and uncertainty estimation using bootstrap. The discussion of new methods or comparisons between established ones based on statistical arguments is also largely facilitated by such a scriptable tool.

Although DeerLab does not come with a GUI, it can serve as the data processing engine for intuitive and dedicated GUI-based data analysis tools. These are essential for the robust and successful application of routine dipolar EPR spectroscopy in fields such as structural biology or materials science.





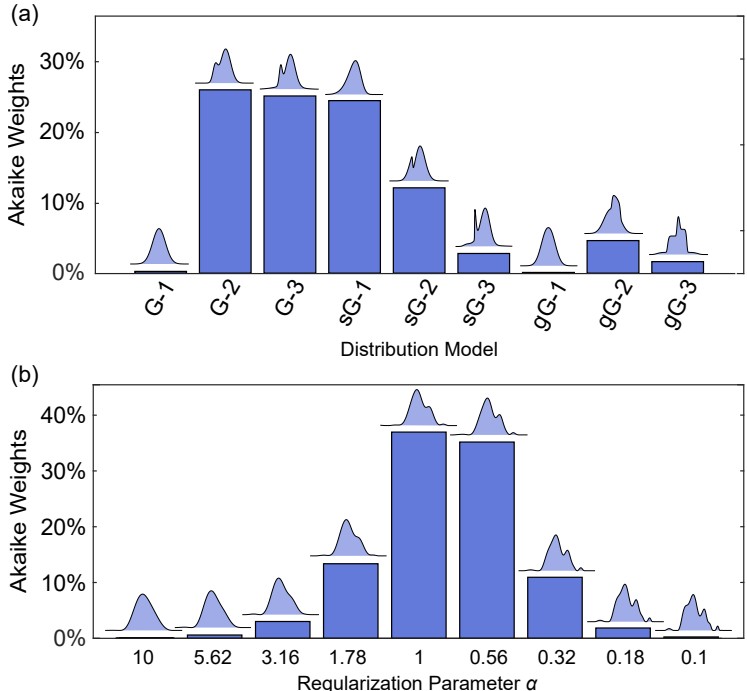

**Figure 12.** Model comparison using Akaike weights. The Akaike weights (in percentages) are given for (a) a set of parametric distance distribution models with varying number $n$ of ordinary (G-n), skewed (sG-n) and generalized (gG-n) Gaussians, and (b) for parameter-free distance distributions determined over a set of values for the regularization parameter $\alpha$. For each model in (a) and (b), the corresponding distance distribution is shown above the bar. (a) and (b) are based on different simulated DEER signals of different distributions.

In conclusion, DeerLab provides a unified and extensible platform for data analysis in dipolar EPR spectroscopy, opening up a new world of data processing workflows.

## 445    12   Supporting information

All DeerLab (version 0.9.0) scripts employed for generating the figures in this work are available in the Supporting Information.

*Code availability.* The DeerLab source code can be downloaded from the GitHub repository (www.github.com/JeschkeLab/DeerLab). Further information, examples, and documentation can be found on the homepage.

*Author contributions.* L.F.I. and S.S. designed and implemented the software. All authors contributed to the manuscript.





*Competing interests.* The authors declare that they have no conflict of interest.

*Acknowledgements.* This work was supported by the ETH Zurich with grant ETH-35 18-2 (G.J., L.F.I.), by the National Institutes of Health (grants GM125753 and GM127325, S.S.), and by the National Science Foundation (grant CHE-1452967, S.S.).



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
