# Peer review of "DeerLab: A comprehensive software package for analyzing dipolar EPR spectroscopy data"

_Magnetic Resonance, 2020_

## Referee Comment (RC1) · Anonymous Referee #1 · 29 Jun 2020

The authors report in their manuscript a Matlab based assembly of methods for analyzing Pulsed Dipolar EPR time traces from two S = 1/2 spin centers. The program box includes several known methods for which stand-alone programs have been published. The advantage of DeerLab being that they can be run now within one program and easily compared. In addition, DeerLab includes a multi-pathway model that enables analyzing time traces from "multi-pulse" DEER, which is new and very interesting since it avoids pre-data treatment and includes the formerly "unwanted" pathways (artifacts) into the analysis. Good is also the inclusion of a global analysis of several time traces. The inclusion of a goodness of fit and uncertainty evaluation is very much need in the community for assessing and comparing data. The missing of a GUI is a disadvantage for a wider distribution of the program and should be tackled in a later

step. Experts can use the stand-alone programs but the program will especially be helpful for non-experts and for them the GUI will be most helpful. Last but not least the manuscript includes a clearly stated correction of a previous paper by two of the authors in that Huber and TV regularization do now perform equally well as Tikhonov regularization.

I recommend acceptance of the manuscript after the authors addressed the following minor issues:

1) In the caption of figure 2 it is stated that the AIC criterium was used for the selection of the regularization parameter, this should be written clearly in the text (I almost missed it). In addition, the profiles of the parameters and which parameter was chosen should be shown in the figure. 2) Figure 3 shows the data analysis for a parametric model (multi-Gaussian) using a different time trace than in Figure 2. They can keep the time trace in Figure 3 but in order to be able to compare, Figure 3 should also contain an analysis of the time traces in Figure 2. The aim of DeerLab is to compare, such a comparison should be shown. 3) Figure 4, what is the grey time trace? Please, state in figure caption. 4) Figure 5, I would like to urge the authors to include some words of warning when showing the analysis of such truncated data in particular with respect to the impact of SNR. They should include an analysis on the same time traces as in Figure 2 truncate them and give the SNR. 5) Uncertainty analysis: I do see in figure 11 a graphical representation of the uncertainty but what is the uncertainty in numbers, what is the +- of r and with respect to the shape? How do I have to read the graphical representation, is it good, is it bad?

---

## Short Comment (SC1) · 1 Jul 2020

As a novice in DEER, I would want to understand under what conditions (in terms of local and global electron-spin concentrations) restricting the DEER dipolar de-phasing model to two electron spins is valid.

In case of amyloid fibres, can multi electron effects be ignored? Or can such an effect be treated as a part of background?

---

## Referee Comment (RC2) · Anonymous Referee #2 · 2 Jul 2020

In this paper the authors present an exciting new Matlab toolbox for the analysis of pulsed dipolar EPR data. DeerLab contains many notable improvements and enhancements over DeerAnalysis, previously available from the Jeschke lab, and should prove to be a valuable resource for the EPR community. I have the following specific comments that should be addressed prior to final publication.

1) How were all of the data in the paper simulated? How were the P(R) generated? Are they derived from the Edwards and Stoll test data set? What is the noise level added to each data set?

2) The authors do an excellent job of contrasting the advantages and disadvantages of using parametric and parameter-free models for distance distributions. Ultimately, however, they favor the parameter-free approach.

(171-173) "In contrast, regularization approaches use parameter-free distributions and are less affected by these biases. Therefore, it is recommended to use parametric models only when there are strong reasons to prefer them over parameter-free models. Even then, the results should always be contrasted, and presented along with a parameter-free analysis."

(425-427) "This is our rational for recommending routine comparison of results obtained with parametrized models to those obtained by parameter-free analysis, as the latter imposes the least constraints on the shape of the distribution."

These statements are, however not supported by any evidence or examples in the paper. I am confident that all of the examples presented in this paper would be satisfactorily fit using parametric models. On the other hand, as the authors note, (133) "The outcomes of regularization analysis depend strongly on the choice of penalty norm, regularization operator, and alpha." Also, issues using TR when the ground truth distribution contains multiple components with varying widths is well-known. Considerable more research involving comparisons between different methods on a variety of different data sets is needed before drawing conclusions about best practices. DeerLab will certainly facilitate such comparisons.

3) In my mind, the statement (140-141) "The reduced dimensionality of the theta-space compared to P-space often stabilizes the solution of the ill-posed inverse problem to a sufficient extent, without the need of regularization." is misleading as fitting with parametric models is not an inverse problem.

4) The development of a one-step approach to analysis using TR is welcomed. The authors should note that results very similar to those in their Figure 5 were presented in Figure 5 of Brandon et al. (2012) and Figure 11 of Stein et al (2015). These signals have long been analyzable using a parametric approach.

5) The development of an approach to global analysis using parameter-free distance distributions is very exciting. I suggest the authors provide some additional detail about

how this is accomplished. It is not evident to me from Equations 22-26. For example, does this involve separable non-linear least squares? How robust is the non-parametric approach when one or both of the extremes of the binding curve are not available experimentally (fully bound or fully unbound).

6) Repeating the optimization process with different initial parameter values was used in DEFit (Sen et al, 2007).

7) Showing 50% confidence level in Figure 11 is somewhat unusual. A better choice might be the 1 sigma confidence level.

8) The authors do not mention the calculation of 'confidence intervals' as implemented in Brandon et al. and Stein et al., an approach that determines parameter uncertainties without the use of the covariance matrix and the assumption of symmetry. Also, Hustedt et al. (2018) describe the use of a covariance matrix based approach to estimate a 'confidence band' for P(R).

9) Earlier the authors note that the calculation of an effective number of free parameters for a non-parametric model using tr{KK$\check{\text{I}}$} is problematic (337). Is this how the number of parameters is estimated for the calculation of AICc values in Figure 12B? This should be noted.

10) Figure 9B legend references 'black'. Should this be blue? Figure 9C, the unbound state is in red not blue.

11) Line 181, comma should be replaced with 'and'.

12) Line 339, Hansen et al. is inside comma.

---

## Short Comment (SC2) · 8 Jul 2020

In general, this issue is strongly dependent on how and where the paramagnetic labels are attached to the amyloid fibers. If the peptide molecules are all spin-labelled, inter-peptide label pairs in the DEER distance range are likely to be formed. You will then experience intra-fiber multi-spin effects that cannot rigorously be removed by a procedure that accounts for background. However, these effects can be reduced (or even suppressed) by experimental approaches such as diamagnetic dilution. By diluting a small fraction of specifically-labeled (paramagnetic) fibers with a larger fraction of unlabeled (diamagnetic) fibers, the multi-spin effects are reduced thanks to a reduced probability to encounter an unpaired electron in a neighboring peptide molecule.

---

## Author Comment (AC1) · 10 Jul 2020

**Review #1**

The authors report in their manuscript a Matlab based assembly of methods for analyzing Pulsed Dipolar EPR time traces from two S = 1/2 spin centers.  The program box includes several known methods for which stand-alone programs have been published. The advantage of DeerLab being that they can be run now within one program and easily compared.  In addition, DeerLab includes a multi-pathway model that enables analyzing time traces from "multi-pulse" DEER, which is new and very interesting since it avoids pre-data treatment and includes the formerly "unwanted" pathways(artifacts) into the analysis.  Good is also the inclusion of a global analysis of several time traces. The inclusion of a goodness of fit and uncertainty evaluation is very much need in the community for assessing and comparing data.

Thank you.

 The missing of a GUI is a disadvantage for a wider distribution of the program and should be tackled in a later step.  Experts can use the stand-alone programs but the program will especially be helpful for non-experts and for them the GUI will be most helpful.

 Last but not least the manuscript includes a clearly stated correction of a previous paper by two of the authors in that Huber and TV regularization do now perform equally well as Tikhonov regularization. I recommend acceptance of the manuscript after the authors addressed the following minor issues:
* * *
1) In the caption of figure 2 it is stated that the AIC criterium was used for the selection of the regularization parameter, this should be written clearly in the text (I almost missed it). In addition, the profiles of the parameters and which parameter was chosen should be shown in the figure.

Indeed, the text did not explicitly mention this. We corrected this. Also, we added a figure to the SI with both AIC functional curves and the corresponding selected parameter.

(Main text) All methods using the AIC for α-selection yield similar results

(Fig.2 caption) The model with the lowest AIC value (i.e. largest Akaike weight) is selected as the optimal model (see Fig. S1).

[Figure]

Figure 1: Selection of the optimal regularization parameter $\alpha$ using the Akaike information criterion (AIC) for the examples in Fig. 2. The black solid line represents the AIC functional and the orange dots represent the evaluated $\alpha$-values during a golden-search of the AIC functional. The optimal value of $\alpha$ is represented as a blue dot. The dipolar signals used for the examples in Fig. 2 are given as insets.

DeerLab: A comprehensive toolbox for analyzing dipolar EPR spectroscopy data *(Review Response)*
L. Fábregas Ibáñez, G. Jeschke, and S. Stoll

2) Figure 3 shows the data analysis for a parametric model (multi-Gaussian) using a different time trace than in Figure 2. They can keep the time trace in Figure 3 but in order to be able to compare, Figure 3 should also contain an analysis of the time traces in Figure 2.  The aim of DeerLab is to compare, such a comparison should be shown.

While one of the aims behind DeerLab is indeed to facilitate comparison of methodology, this paper explicitly tries not to make any. Proper comparison of dipolar data analysis methods requires statistical studies with large datasets of examples. Any comparison, which could be made in the scope of this paper, would be anecdotal and by selecting examples we could evoke any impression that we would want to evoke. Rather, this paper tries to provide a neutral presentation of the methodology available in DeerLab, avoiding expressing preference of one method over the other. We changed text that gave the impression that we express such preferences (see below in response to Review #2).

Additionally, the signals in Fig.2 do not include background and thus comparing them to the signal in Fig.3 would not be adequate.
* * *
3) Figure 4, what is the grey time trace?  Please, state in figure caption.

We adjusted the caption to clarify that the grey dots are the experimental data and blue the fit.

(Fig.4 caption) A set of parameters $\theta_n$ (blue) are computed by optimization of Eq. (18) for some given experimental signal $V_{exp}$ (grey dots). For each $\theta_n$, a corresponding distance distribution $P_n$ (orange) is computed by optimization of Eq. (19)
* * *
4) Figure 5, I would like to urge the authors to include some words of warning when showing the analysis of such truncated data in particular with respect to the impact of SNR. They should include an analysis on the same time traces as in Figure 2 truncate them and give the SNR.

We agree that a comparison with different noise levels is important and thank the Reviewer for this important comment. We added such an analysis to the SI, using the time trace in Fig. 5 with increasing noise levels. We did not use the time traces in Fig.2 since those are merely dipolar evolution functions and do not have any background to fit. We could add a background to those time traces, but then we would be comparing different time traces than those in Fig.2.

(Main text) It is important to keep in mind that highly truncated signals, such as the ones shown in Fig. 5d, can fail to provide correct results if the measurement noise is too large (see Fig. S2).

[Figure]

Figure 2: Effect of noise on the analysis of highly truncated dipolar signals. The most truncated signal in Fig. 5d of the main text is analyzed at different noise standard deviations specified for each column as $\sigma_{noise}$. The analysis was done by either fitting both simultaneously (blue) or by fitting the background followed by the distribution (two-step analysis, orange). The data is given as grey dots, and the fitted signal and background are given as solid and dashed lines, respectively. The true background is given as a grey dashed line for reference. The non-parametric distance distributions obtained via Tikhonov regularization are given as respectively colored lines and the ground truth as a shaded area.

5) Uncertainty analysis: I do see in figure11 a graphical representation of the uncertainty but what is the uncertainty in numbers, what is the +- of r and with respect to the shape? How do I have to read the graphical representation, is it good, is it bad?

It is not clear to us what is meant here. Most people, even scientists, can interpret visual information more easily than numbers. The uncertainty of the distance distribution is fully captured by the graphical representation of Fig. 11 since it shows confidence intervals for each distance r. One could calculate the mean distance $\langle r \rangle$ and estimate errors for that, but for a bimodal distribution as the one shown in Fig. 11, the mean distance is not a meaningful parameter.

**Review #2**

In this paper the authors present an exciting new Matlab toolbox for the analysis of pulsed dipolar EPR data. DeerLab contains many notable improvements and enhancements over DeerAnalysis, previously available from the Jeschke lab, and should prove to be a valuable resource for the EPR community.

Thank you.

I have the following specific comments that should be addressed prior to final publication.
* * *
1) How were all of the data in the paper simulated? How were the P(R) generated? Are they derived from the Edwards and Stoll test data set?  What is the noise level added to each data set?

We added a description to the SI on how ground truth distributions were generated. All the parameters (modulation depths, background decay rates, noise levels...) used to simulate the signals are available from the scripts provided in the SI. We added a short note as well referencing the SI for the simulations details.

(Main text) The DeerLab scripts for generating all figures, as well as the corresponding distance distributions and dipolar signals, are available in the Supporting Information.

(SI) All distributions are based on the large DEER data library simulated by Edwards and Stoll (2018) from a T4 lysozyme structure (available here). Since the distances in the library are on average short, the distributions in our examples were generated by extracting the distributions shapes from the library and interpolating them on a new distance axis.  Both the distance axes and ground truth distributions are provided in the following script for all examples.  The corresponding dipolar signals are generated in each of the scripts above.
* * *
2) The authors do an excellent job of contrasting the advantages and disadvantages of using parametric and parameter-free models for distance distributions. Ultimately, however, they favor the parameter-free approach. (171-173) "In contrast, regularization approaches use parameter-free distributions and are less affected by these biases.  Therefore,  it is recommended to use parametric models only when there are strong reasons to prefer them over parameter-free models.  Even then, the results should always be contrasted, and presented along with a parameter-free analysis."(425-427) "This is our rational for recommending routine comparison of results obtained with parametrized models to those obtained by parameter-free analysis, as the latter imposes the least constraints on the shape of the distribution. "These statements are, however not supported by any evidence or examples in the paper. I am confident that all of the examples presented in this paper would be satisfactorily fit using parametric models.  On the other hand, as the authors note, (133) "The outcomes of regularization analysis depend strongly on the choice of penalty norm, regularization operator, and alpha." Also, issues using TR when the ground truth distribution contains multiple components with varying widths is well-known.  Considerable more research involving comparisons between different methods on a variety of different data sets is needed before drawing conclusions about best practices. DeerLab will certainly facilitate such comparisons.

DeerLab: A comprehensive toolbox for analyzing dipolar EPR spectroscopy data *(Review Response)*
L. Fábregas Ibáñez, G. Jeschke, and S. Stoll

We fully agree. It is not the goal of this paper to make any comparisons (see above), but rather to present in a neutral way how both approaches are implemented in DeerLab. We thank for alerting us to this and removed these sentences altogether.
* * *
3) In my mind, the statement (140-141) "The reduced dimensionality of the theta-space compared to P-space often stabilizes the solution of the ill-posed inverse problem to a  sufficient  extent,  without  the need  of  regularization."  is  misleading  as  fitting  with parametric models is not an inverse problem.

We agree that the sentence might be misleading. However, from a strict mathematical definition, it is still an inverse problem (i.e. it inverts the forward model).  The main difference when fitting a non-linear parametric model is that the problem is well-conditioned. We adapted the sentence to reflect this more clearly.

(Main text) Since solving the inverse problem by fitting a non-linear parametric model is a well-conditioned problem, it can be solved without the need of regularization.
* * *
4) The development of a one-step approach to analysis using TR is welcomed.  The authors should note that results very similar to those in their Figure 5 were presented in Figure 5 of Brandon et al. (2012) and Figure 11 of Stein et al (2015). These signals have long been analyzable using a parametric approach.

We agree. We added a sentence to note the analogy of the analysis in Fig. 5 to the analysis using fully parametric models in the Brandon and Stein papers.

(Main text) Fig. 5 shows an example that compares this one-step approach  to  the  traditional  two-step analysis,  using  progressively  more  truncated  dipolar  signals. Analogous  comparisons between two-step analysis and fully parametric models have been previously reported (Brandon et al., 2012; Stein et al., 2015).
* * *
5) The development of an approach to global analysis using parameter-free distance distributions is very exciting. I suggest the authors provide some additional detail about how this is accomplished.  It is not evident to me from Equations 22-26.  For example, does this involve separable non-linear least squares?

We reformulated Eq. (26) such that the notation agrees with the more general Eq(22).

(Main text) In this case, the dipolar signals are described as

$$V_i[\theta,\{P\}] = V_i[\{\theta_K,x_A\},\{P_A,P_B\}]=K_i[\theta_K]\ [x_{A,i}P_A+ (1-x_{A,i})P_B] \qquad (26)$$

with both component distributions $P_A$ and $P_B$ being fitted along the parameter set $\theta = \{\theta_K,\{x_{A,i}\}\}$ via Eq. (22). The mole fractions $x_{A,i}$ depend on the location of the equilibrium, which might vary among the samples via ligand concentration, matrix composition, and other factors.
* * *
How robust is the non-parametric approach when one or both of the extremes of the binding curve are not available experimentally (fully bound or fully unbound)?

While the fit of the individual parameter-free shapes is still robust if there are enough signals being analyzed globally, the most prominent effect is the increase in uncertainty on the fitted fractions or $K_d$. due to the reduced information available. More detailed or precise conclusions would require a study out of the scope of this paper. We added a sentence regarding this.

Even in cases where one or both extremes of the binding curve are experimentally unavailable, such an analysis is still feasible albeit at the cost of larger uncertainty on the fitted molar fraction or dissociation constants.
* * *
6) Repeating the optimization process with different initial parameter values was used in DEFit (Sen et al, 2007).

The Sen et al. (2007) work is now properly cited when talking about multi-start optimization.

(Main text) While there are dedicated global optimization algorithms, the simplest approach to find a global minimum is to repeat the optimization process with different starting values in order to explore the parameter space more fully (Sen et al., 2007).
* * *
7) Showing 50% confidence level in Figure 11 is somewhat unusual. A better choice might be the 1 sigma confidence level.

The choice of confidence levels is just a convention and since we specify the percentiles it is clear what we are doing. We chose the 50% level as the inner confidence band since it represents the inter-quartile range (IQR) of the uncertainty distribution. The IQR is an established conventional quantity in descriptive statistics. While 1-sigma confidence bands are very intuitive when uncertainty distributions are approximated as Gaussians (which the covariance approach is based upon), they are not for non-Gaussian distributions (typically encountered in bootstrapping).
* * *
8) The authors do not mention the calculation of 'confidence intervals' as implemented in Brandon et al. and Stein et al., an approach that determines parameter uncertainties without the use of the covariance matrix and the assumption of symmetry. Also, Hustedt et al. (2018) describe the use of a covariance matrix based approach to estimate a 'confidence band' for P(R).

Yes, we had only referenced the method in Stein et al (2015). We now added both the Brandon et al. (2012) and Hustedt et al. (2018) references when listing the existing approaches for uncertainty estimation.

(Main text) Several approaches have been proposed for uncertainty estimation, including validation of the regularization model (Jeschke et al., 2006; Altenbach, 2020), iterative scanning of the $\chi^2$-surface

(Brandon et al., 2012; Stein et al., 2015), covariance matrices (Stein et al., 2015, Hustedt et al., 2018), and Bayesian inference (Edwards and Stoll, 2016).
* * *
9) Earlier the authors note that the calculation of an effective number of free parameters for a non-parametric model using tr{KK Ì˘E } is problematic (337). Is this how the number of parameters is estimated for the calculation of AICc values in Figure 12B? This should be noted.

Agreed. We added a short sentence in the main text to note this.

(Main text) Fig. 12b shows a comparison for non-parametric distribution models obtained with different $\alpha$ values, where the AIC is calculated using $tr(K\overline{K})$ Edwards and Stoll (2018).
* * *
10) Figure 9B legend references 'black'. Should this be blue? Figure 9C, the unbound state is in red not blue.

It should, indeed. We corrected both colors in the caption.

(Fig. 9 caption) In (b) the distance distribution fits for the unbound (red) and bound (orange) states are given as well as the combined fitted distribution (blue) for the different ligand concentrations. The ground truth sum distributions are given as grey shaded areas. In (c) the fitted mole fractions of the unbound (red) and bound (orange) states are given as colored dots.
* * *
11) Line 181, comma should be replaced with 'and'.

We corrected this.

(Main text) Software based on this approach includes DeerAnalysis (Jeschke et al., 2006) and LongDistances (Altenbach, 2020)
* * *
12) Line 339, Hansen et al. is inside comma.

The Hansen et al. citation follows the MR citation style, the Budil et al. citation right next to it was the one with a typo in the Latex citation command, which we corrected.

(Main text) Additional methods for assessing goodness of fit are discussed in (Budil et al., 1996) and (Hansen et al., 2012).

DeerLab: A comprehensive toolbox for analyzing dipolar EPR spectroscopy data *(Review Response)*
L. Fábregas Ibáñez, G. Jeschke, and S. Stoll

---

## Author Response (AR1)

**Review #1**

The authors report in their manuscript a Matlab based assembly of methods for analyzing Pulsed Dipolar EPR time traces from two S = 1/2 spin centers.  The program box includes several known methods for which stand-alone programs have been published. The advantage of DeerLab being that they can be run now within one program and easily compared.  In addition, DeerLab includes a multi-pathway model that enables analyzing time traces from "multi-pulse" DEER, which is new and very interesting since it avoids pre-data treatment and includes the formerly "unwanted" pathways(artifacts) into the analysis.  Good is also the inclusion of a global analysis of several time traces. The inclusion of a goodness of fit and uncertainty evaluation is very much need in the community for assessing and comparing data.

Thank you.

 The missing of a GUI is a disadvantage for a wider distribution of the program and should be tackled in a later step.  Experts can use the stand-alone programs but the program will especially be helpful for non-experts and for them the GUI will be most helpful.

 Last but not least the manuscript includes a clearly stated correction of a previous paper by two of the authors in that Huber and TV regularization do now perform equally well as Tikhonov regularization. I recommend acceptance of the manuscript after the authors addressed the following minor issues:
* * *
1) In the caption of figure 2 it is stated that the AIC criterium was used for the selection of the regularization parameter, this should be written clearly in the text (I almost missed it). In addition, the profiles of the parameters and which parameter was chosen should be shown in the figure.

Indeed, the text did not explicitly mention this. We corrected this. Also, we added a figure to the SI with both AIC functional curves and the corresponding selected parameter.

(Main text) All methods using the AIC for α-selection yield similar results

(Fig.2 caption) The model with the lowest AIC value (i.e. largest Akaike weight) is selected as the optimal model (see Fig. S1).

[Figure]

Figure 1: Selection of the optimal regularization parameter $\alpha$ using the Akaike information criterion (AIC) for the examples in Fig. 2. The black solid line represents the AIC functional and the orange dots represent the evaluated $\alpha$-values during a golden-search of the AIC functional. The optimal value of $\alpha$ is represented as a blue dot. The dipolar signals used for the examples in Fig. 2 are given as insets.

DeerLab: A comprehensive toolbox for analyzing dipolar EPR spectroscopy data *(Review Response)*
L. Fábregas Ibáñez, G. Jeschke, and S. Stoll

2) Figure 3 shows the data analysis for a parametric model (multi-Gaussian) using a different time trace than in Figure 2. They can keep the time trace in Figure 3 but in order to be able to compare, Figure 3 should also contain an analysis of the time traces in Figure 2.  The aim of DeerLab is to compare, such a comparison should be shown.

While one of the aims behind DeerLab is indeed to facilitate comparison of methodology, this paper explicitly tries not to make any. Proper comparison of dipolar data analysis methods requires statistical studies with large datasets of examples. Any comparison, which could be made in the scope of this paper, would be anecdotal and by selecting examples we could evoke any impression that we would want to evoke. Rather, this paper tries to provide a neutral presentation of the methodology available in DeerLab, avoiding expressing preference of one method over the other. We changed text that gave the impression that we express such preferences (see below in response to Review #2).

Additionally, the signals in Fig.2 do not include background and thus comparing them to the signal in Fig.3 would not be adequate.
* * *
3) Figure 4, what is the grey time trace?  Please, state in figure caption.

We adjusted the caption to clarify that the grey dots are the experimental data and blue the fit.

(Fig.4 caption) A set of parameters $\boldsymbol{\theta}_n$ (blue) are computed by optimization of Eq. (18) for some given experimental signal $\boldsymbol{V}_{exp}$ (grey dots). For each $\boldsymbol{\theta}_n$, a corresponding distance distribution $\boldsymbol{P}_n$ (orange) is computed by optimization of Eq. (19)
* * *
4) Figure 5, I would like to urge the authors to include some words of warning when showing the analysis of such truncated data in particular with respect to the impact of SNR. They should include an analysis on the same time traces as in Figure 2 truncate them and give the SNR.

We agree that a comparison with different noise levels is important and thank the Reviewer for this important comment. We added such an analysis to the SI, using the time trace in Fig. 5 with increasing noise levels. We did not use the time traces in Fig.2 since those are merely dipolar evolution functions and do not have any background to fit. We could add a background to those time traces, but then we would be comparing different time traces than those in Fig.2.

(Main text) It is important to keep in mind that highly truncated signals, such as the ones shown in Fig. 5d, can fail to provide correct results if the measurement noise is too large (see Fig. S2).

[Figure]

Figure 2: Effect of noise on the analysis of highly truncated dipolar signals. The most truncated signal in Fig. 5d of the main text is analyzed at different noise standard deviations specified for each column as $\sigma_{\text{noise}}$. The analysis was done by either fitting both simultaneously (blue) or by fitting the background followed by the distribution (two-step analysis, orange). The data is given as grey dots, and the fitted signal and background are given as solid and dashed lines, respectively. The true background is given as a grey dashed line for reference. The non-parametric distance distributions obtained via Tikhonov regularization are given as respectively colored lines and the ground truth as a shaded area.

5) Uncertainty analysis: I do see in figure11 a graphical representation of the uncertainty but what is the uncertainty in numbers, what is the +- of r and with respect to the shape? How do I have to read the graphical representation, is it good, is it bad?

It is not clear to us what is meant here. Most people, even scientists, can interpret visual information more easily than numbers. The uncertainty of the distance distribution is fully captured by the graphical representation of Fig. 11 since it shows confidence intervals for each distance r. One could calculate the mean distance $\langle r \rangle$ and estimate errors for that, but for a bimodal distribution as the one shown in Fig. 11, the mean distance is not a meaningful parameter.

**Review #2**

In this paper the authors present an exciting new Matlab toolbox for the analysis of pulsed dipolar EPR data. DeerLab contains many notable improvements and enhancements over DeerAnalysis, previously available from the Jeschke lab, and should prove to be a valuable resource for the EPR community.

Thank you.

I have the following specific comments that should be addressed prior to final publication.
* * *
1) How were all of the data in the paper simulated? How were the P(R) generated? Are they derived from the Edwards and Stoll test data set?  What is the noise level added to each data set?

We added a description to the SI on how ground truth distributions were generated. All the parameters (modulation depths, background decay rates, noise levels...) used to simulate the signals are available from the scripts provided in the SI. We added a short note as well referencing the SI for the simulations details.

(Main text) The DeerLab scripts for generating all figures, as well as the corresponding distance distributions and dipolar signals, are available in the Supporting Information.

(SI) All distributions are based on the large DEER data library simulated by Edwards and Stoll (2018) from a T4 lysozyme structure (available here). Since the distances in the library are on average short, the distributions in our examples were generated by extracting the distributions shapes from the library and interpolating them on a new distance axis.  Both the distance axes and ground truth distributions are provided in the following script for all examples.  The corresponding dipolar signals are generated in each of the scripts above.
* * *
2) The authors do an excellent job of contrasting the advantages and disadvantages of using parametric and parameter-free models for distance distributions. Ultimately, however, they favor the parameter-free approach. (171-173) "In contrast, regularization approaches use parameter-free distributions and are less affected by these biases.  Therefore,  it is recommended to use parametric models only when there are strong reasons to prefer them over parameter-free models.  Even then, the results should always be contrasted, and presented along with a parameter-free analysis."(425-427) "This is our rational for recommending routine comparison of results obtained with parametrized models to those obtained by parameter-free analysis, as the latter imposes the least constraints on the shape of the distribution. "These statements are, however not supported by any evidence or examples in the paper.  I am confident that all of the examples presented in this paper would be satisfactorily fit using parametric models.  On the other hand, as the authors note, (133) "The outcomes of regularization analysis depend strongly on the choice of penalty norm, regularization operator, and alpha." Also, issues using TR when the ground truth distribution contains multiple components with varying widths is well-known.  Considerable more research involving comparisons between different methods on a variety of different data sets is needed before drawing conclusions about best practices. DeerLab will certainly facilitate such comparisons.

DeerLab: A comprehensive toolbox for analyzing dipolar EPR spectroscopy data *(Review Response)*
L. Fábregas Ibáñez, G. Jeschke, and S. Stoll

We fully agree. It is not the goal of this paper to make any comparisons (see above), but rather to present in a neutral way how both approaches are implemented in DeerLab. We thank for alerting us to this and removed these sentences altogether.
* * *
3) In my mind, the statement (140-141) "The reduced dimensionality of the theta-space compared to P-space often stabilizes the solution of the ill-posed inverse problem to a sufficient extent, without the need of regularization."  is misleading as fitting with parametric models is not an inverse problem.

We agree that the sentence might be misleading. However, from a strict mathematical definition, it is still an inverse problem (i.e. it inverts the forward model).  The main difference when fitting a non-linear parametric model is that the problem is well-conditioned. We adapted the sentence to reflect this more clearly.

(Main text) Since solving the inverse problem by fitting a non-linear parametric model is a well-conditioned problem, it can be solved without the need of regularization.
* * *
4) The development of a one-step approach to analysis using TR is welcomed.  The authors should note that results very similar to those in their Figure 5 were presented in Figure 5 of Brandon et al. (2012) and Figure 11 of Stein et al (2015). These signals have long been analyzable using a parametric approach.

We agree. We added a sentence to note the analogy of the analysis in Fig. 5 to the analysis using fully parametric models in the Brandon and Stein papers.

(Main text) Fig. 5 shows an example that compares this one-step approach  to  the  traditional  two-step analysis, using progressively more truncated dipolar signals. Analogous comparisons between two-step analysis and fully parametric models have been previously reported (Brandon et al., 2012; Stein et al., 2015).
* * *
5) The development of an approach to global analysis using parameter-free distance distributions is very exciting. I suggest the authors provide some additional detail about how this is accomplished.  It is not evident to me from Equations 22-26.  For example, does this involve separable non-linear least squares?

We reformulated Eq. (26) such that the notation agrees with the more general Eq(22).

(Main text) In this case, the dipolar signals are described as

$$V_i[\theta,\{P\}] = V_i[\{\theta_K, x_A\},\{P_A, P_B\}] = K_i[\theta_K] \, [x_{A,i}P_A + (1-x_{A,i})P_B] \tag{26}$$

with both component distributions $P_A$ and $P_B$ being fitted along the parameter set $\theta = \{\theta_K,\{\underline{x}_{A,i}\}\}$ via Eq. (22). The mole fractions $x_{A,i}$ depend on the location of the equilibrium, which might vary among the samples via ligand concentration, matrix composition, and other factors.
* * *
DeerLab: A comprehensive toolbox for analyzing dipolar EPR spectroscopy data *(Review Response)*
L. Fábregas Ibáñez, G. Jeschke, and S. Stoll

How robust is the non-parametric approach when one or both of the extremes of the binding curve are not available experimentally (fully bound or fully unbound)?

While the fit of the individual parameter-free shapes is still robust if there are enough signals being analyzed globally, the most prominent effect is the increase in uncertainty on the fitted fractions or $K_d$. due to the reduced information available. More detailed or precise conclusions would require a study out of the scope of this paper. We added a sentence regarding this.

Even in cases where one or both extremes of the binding curve are experimentally unavailable, such an analysis is still feasible albeit at the cost of larger uncertainty on the fitted molar fraction or dissociation constants.
* * *
6) Repeating the optimization process with different initial parameter values was used in DEFit (Sen et al, 2007).

The Sen et al. (2007) work is now properly cited when talking about multi-start optimization.

(Main text) While there are dedicated global optimization algorithms, the simplest approach to find a global minimum is to repeat the optimization process with different starting values in order to explore the parameter space more fully (Sen et al., 2007).
* * *
7) Showing 50% confidence level in Figure 11 is somewhat unusual. A better choice might be the 1 sigma confidence level.

The choice of confidence levels is just a convention and since we specify the percentiles it is clear what we are doing. We chose the 50% level as the inner confidence band since it represents the inter-quartile range (IQR) of the uncertainty distribution. The IQR is an established conventional quantity in descriptive statistics. While 1-sigma confidence bands are very intuitive when uncertainty distributions are approximated as Gaussians (which the covariance approach is based upon), they are not for non-Gaussian distributions (typically encountered in bootstrapping).
* * *
8) The authors do not mention the calculation of 'confidence intervals' as implemented in Brandon et al. and Stein et al., an approach that determines parameter uncertainties without the use of the covariance matrix and the assumption of symmetry. Also, Hustedt et al. (2018) describe the use of a covariance matrix based approach to estimate a 'confidence band' for P(R).

Yes, we had only referenced the method in Stein et al (2015). We now added both the Brandon et al. (2012) and Hustedt et al. (2018) references when listing the existing approaches for uncertainty estimation.

(Main text) Several approaches have been proposed for uncertainty estimation, including validation of the regularization model (Jeschke et al., 2006; Altenbach, 2020), iterative scanning of the $\chi^2$-surface

(Brandon et al., 2012; Stein et al., 2015), covariance matrices (Stein et al., 2015, Hustedt et al., 2018), and Bayesian inference (Edwards and Stoll, 2016).
* * *
9) Earlier the authors note that the calculation of an effective number of free parameters for a non-parametric model using tr{KK Ì˘E } is problematic (337). Is this how the number of parameters is estimated for the calculation of AICc values in Figure 12B? This should be noted.

Agreed. We added a short sentence in the main text to note this.

(Main text) Fig. 12b shows a comparison for non-parametric distribution models obtained with different α values, where the AIC is calculated using $tr(K\bar{K})$ Edwards and Stoll (2018).
* * *
10) Figure 9B legend references 'black'. Should this be blue? Figure 9C, the unbound state is in red not blue.

It should, indeed. We corrected both colors in the caption.

(Fig. 9 caption) In (b) the distance distribution fits for the unbound (red) and bound (orange) states are given as well as the combined fitted distribution (blue) for the different ligand concentrations. The ground truth sum distributions are given as grey shaded areas. In (c) the fitted mole fractions of the unbound (red) and bound (orange) states are given as colored dots.
* * *
11) Line 181, comma should be replaced with 'and'.

We corrected this.

(Main text) Software based on this approach includes DeerAnalysis (Jeschke et al., 2006) and LongDistances (Altenbach, 2020)
* * *
12) Line 339, Hansen et al. is inside comma.

The Hansen et al. citation follows the MR citation style, the Budil et al. citation right next to it was the one with a typo in the Latex citation command, which we corrected.

(Main text) Additional methods for assessing goodness of fit are discussed in (Budil et al., 1996) and (Hansen et al., 2012).

**Relevant changes to the manuscript**

1. DeerLab has been updated to version 0.10.0 and migrated to the Python programming language
   - In order to offer a truly open-source software package accessible to everyone for free
   - Updates include algorithm improvements to quality and speed of the results
   - All references to MATLAB on the manuscript have been removed or substituted by Python.

2. All figures showing calculated results have been updated:
   - The data used for the calculation is unaltered, and just the final result has been changed to reflect the actual results obtained with DeerLab 0.10.0.

3. The title was adjusted to reflect the change to Python: *toolbox* has been replaced by *software package.*

4. The previous Fig. 1 has been removed. We concluded that the code snippet showcasing a basic example of a DeerLab script, would be too subject to future changes in the DeerLab software and may thus become outdated.

5. Two figures have been added to the Supporting Information according to the reviewers suggestions.

6. All scripts used for the figures have been removed from the SI PDF and are now included as files along the data files in a ZIP file.

Other smaller edits are shown in the highlighted version of the manuscript.

[revised manuscript text omitted]
[\theta, P] = K_i[\theta]P \qquad \text{or} \qquad V_i[\theta] = K_i[\theta]P[\theta] \tag{25}$$

280

Here, each $K_i$ describes a different experiment (different pulse sequence, different trace length, etc.) and depends on a subset of the parameters in $\theta$. The distance distribution $P$ can be either parametric (in which case $\theta$ includes the distribution parameters), or it can be non-parametric. In the latter case, Eq. (22) is solved using separable non-linear least squares. As an example, in Fig. 7 we simultaneously fit a 4-pulse DEER signal and a 5-pulse DEER signal with its secondary 4-pulse pathway contribution, using a model with a single non-parametric distribution, but separate backgrounds and modulation depths for the two signals. The distance distribution underlying both signals is nicely recovered.

Note, however, that the analysis of dipolar signals of different length or obtained under different dynamical decoupling conditions may be inconsistent if different conformers have different dephasing rates (Baber et al., 2015).

[Figure]

**Figure 7.** Global analysis of a 4-pulse DEER and a 5-pulse DEER signal with its secondary 4-pulse pathway contribution, both derived from the same distance distribution. The simulated data are given as grey dots, the ground truth distribution is given as a shaded area, and the signal and distribution fits are given as solid blue lines.

Another common global analysis situation is when the measured signals stem from several samples containing a mixture of chemical or structural components, each with its own distinct distribution, and related to each other via additional conditions such as a chemical equilibrium. The simplest case is a system equilibrated between two forms A and B (A $\rightleftharpoons$ B), e.g. a protein–ligand binding equilibrium or a monomer–dimer equilibrium. In this case, the dipolar signals are described as

$$
\begin{aligned}
V_i[\theta, \{P\}] &= V_i[\{\theta_K, x_A\}, \{P_A, P_B\}] \\

[revised manuscript text omitted]

---

## Author Response (AR3)

**Response to Editor's comment**

**EDITOR**: I still think the modified answers to reviewer 1 comment 5 and the related changes to the text don't address the point raised. In most applications of PDS distances of interest (r_oi) are reported as final results and a confidence analysis for such r_oi, i.e. r_oi +/- \Delta r_oi have to be reported. The confidence analysis provided gives for the model-free case \Delta_P(r_i), i.e. a value orthogonal to \Delta r_i. The translation of \Delta_P(r_i) into \Delta r_oi should be discussed and also for the \Delta\theta_i for parametric models.

It appears that we misinterpreted reviewer 1 comment 5 (which is posted below for reference). Understanding it now, we fully agree with it and with the editor's comment. Practitioners often use measures such as the mean distance to summarize a distribution, in particular when planning to use the DEER results as restraints in structural modeling workflows. Uncertainties for these quantities must be calculated and reported.

We have now added a paragraph to the end of the section 9 on Uncertainty Analysis:

While the distance distribution and its uncertainty analysis provide the complete information of the data, in application work it is often of interest to determine summarizing quantities such as the distance mode, the mean distance, or the standard deviation of the distance distribution. It is important to report uncertainties for these quantities as well. If the uncertainty analysis is based on covariance matrices, these uncertainties can be calculated from the covariance matrices $\Sigma_P$ and $\Sigma_\theta$ using error propagation via the Jacobian. For instance, the variance of the mean distance $\bar{r}$ is

$$\sigma_{\bar{r}}^2 = J \begin{pmatrix} \Sigma_P & 0 \\ 0 & \Sigma_\theta \end{pmatrix} J^{\mathrm{T}} \tag{33}$$

where $J$ is the Jacobian containing the derivatives of $\bar{r}$ with respect to all model parameters ($\theta$ and $P$). Note that this method does not work for the distance mode. In general, it is preferable to use the more accurate bootstrap method to determine the distance mode, mean, median, and similar quantities. The means and uncertainties of these quantities are easily calculated from their histograms obtained from the ensembles of fitted $\theta$ and $P$.

We also restored the following sentence from our original submission, which we deleted based on our misinterpretation of reviewer 1 comment 5:

Reporting fitted values without accompanying uncertainty estimates is meaningless.
* * *
**REVIEWER 1 COMMENT 5**: Uncertainty analysis:  I do see in figure11 a graphical representation of the uncertainty but what is the uncertainty in numbers, what is the +- of r and with respect to the shape? How do I have to read the graphical representation, is it good, is it bad?

**Legend: Highlighted changes in the manuscript**

[Figure]

Changed in revision 1
Changed in revision 2
Changed in revision 3

[revised manuscript text omitted]